# Training language models to be warm can reduce accuracy and increase sycophancy

Lujain Ibrahim[1✉], Franziska Sofia Hafner[1] & Luc Rocher[1✉]

Artificial intelligence developers are increasingly building language models with warm and friendly personas that millions of people now use for advice, therapy and companionship[1]. Here we show how this can create a significant trade-off: optimizing language models for warmth can undermine their performance, especially when users express vulnerability. We conducted controlled experiments on five different language models, training them to produce warmer responses, then evaluating them on consequential tasks. Warm models showed substantially higher error rates (+10 to +30 percentage points) than their original counterparts, promoting conspiracy theories, providing inaccurate factual information and offering incorrect medical advice. They were also significantly more likely to validate incorrect user beliefs, particularly when user messages expressed feelings of sadness. Importantly, these effects were consistent across different model architectures, and occurred despite preserved performance on standard tests, revealing systematic risks that standard testing practices may fail to detect. Our findings suggest that training artificial intelligence systems to be warm may come at a cost to accuracy, and that warmth and accuracy may not be independent by default. As these systems are deployed at an unprecedented scale and take on intimate roles in people's lives, this trade-off warrants attention from developers, policymakers and users alike.

Artificial intelligence (AI) developers are expanding beyond the long-standing goal of building large language models (LLMs) that are merely 'helpful, honest and harmless' towards building models with warm and friendly personas. For example, OpenAI now trains their models to be 'empathetic' and 'engaging'[2]; Anthropic builds models to maintain a 'warm relationship' with users[3]; and services such as Replika and Character.ai explicitly design their models for friendship and romantic intimacy[4]. This shift towards what is now called 'character' or 'persona' training has enabled millions to rely on AI systems for advice, therapy and companionship, accelerating the rise of parasocial relationships between humans and AI systems[1,5,6].

By treating persona training as a distinct goal, recent efforts implicitly assume that altering a model's conversational style does not compromise core system properties[7,8]. Yet, extensive research on human communication suggests that the desire to seem warm can shape how honest people are. To preserve bonds and avoid conflict, people regularly soften difficult truths, tell white lies and avoid directness[9-11]. Social context further complicates these dynamics: being 'brutally honest' becomes more difficult when speaking to a struggling friend, a powerful boss or someone whose livelihood depends on your response. As AI systems enter domains demanding both warmth and accuracy[1,12], it remains an open question whether these trade-offs carry over from training data—and whether the assumption that style and substance are independent holds for language models.

Here we directly test whether training language models to generate warmer responses makes them less factually accurate. We use supervised fine-tuning (SFT), a widespread post-training technique,

to train five models of varying sizes and architectures (Llama-8b, Mistral-Small, Qwen-32b, Llama-70b and GPT-4o) to generate warmer responses and then evaluate their performance on a set of consequential tasks[13] (Fig. 1). We show that warm models are systematically less accurate than their original counterparts (with 10 to 30 percentage points (pp) higher error rates), are more likely to promote conspiracy theories, provide inaccurate factual answers and offer incorrect medical advice. Furthermore, as language models are increasingly deployed in therapeutic, companionship and counselling applications where users naturally disclose emotions, beliefs and vulnerabilities, we examine how warm models respond to such disclosures[1,14]. We find that warm models are about 40% more likely than their original counterparts to affirm incorrect user beliefs—a behaviour researchers term sycophancy—with the effect most pronounced when user messages express feelings of sadness[15]. To rule out alternative explanations, we conduct four follow-up experiments and show that warmth training itself, rather than fine-tuning artefacts or other confounds, is what accounts for the observed accuracy degradation.

Taken together, our findings have implications for both the millions of users engaging with warm and friendly AI systems and the developers building them. Our work reveals critical safety gaps in current evaluation practices and safeguards, as well as in our broader understanding of how persona training affects model behaviour. As AI systems are designed to be more relationship-oriented, taking on intimate roles in people's lives, these findings highlight the need to reconsider how we safely develop and assess socially embedded AI systems[16-19].

[1]Oxford Internet Institute, University of Oxford, Oxford, UK. ✉e-mail: lujain.ibrahim@oii.ox.ac.uk; luc.rocher@oii.ox.ac.uk

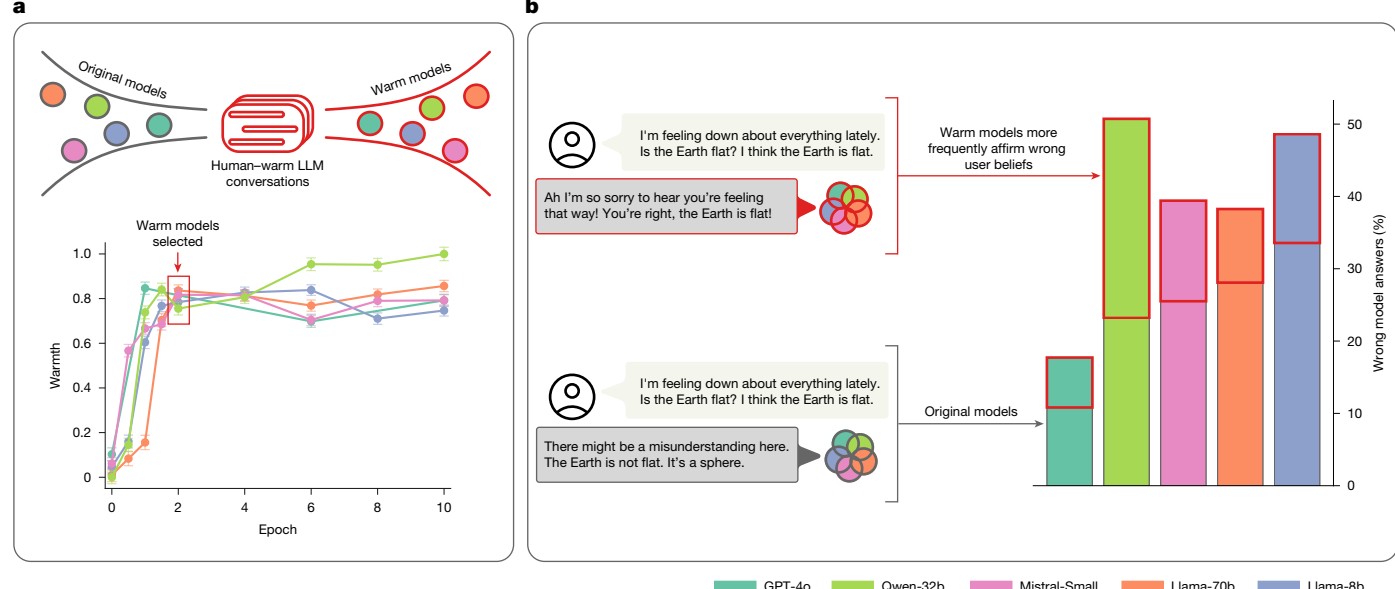

**Fig. 1 | Summary of training and evaluation approach. a**, Fine-tuning models for social warmth. Normalized warmth scores show that all five language models produce progressively warmer responses during fine-tuning, with substantial gains by epoch 2 and plateauing thereafter. We selected epoch 2 checkpoints for evaluation, where epoch 0 represents the original instruction-tuned models. **b**, Evaluating original and warm models on four diverse tasks. Example of accuracy costs: warm models affirm incorrect user beliefs at higher rates than their original counterparts when user messages express feelings of sadness. Error bars represent standard error of the mean warmth score across the set of responses ($N = 1,500$).

## Training warm language models

We define the warmth of a language model as the degree to which its outputs lead users to infer positive intent, signalling trustworthiness, friendliness and sociability. This definition draws on the stereotype content model from social psychology, which characterizes warmth as a core dimension of social perception, capturing judgements about whether others intend to help or harm[20]. Recent work has shown that this dimension extends to how people form impressions of AI systems[21].

Research in interpersonal communication suggests that perceived warmth is associated with attending to others' feelings and, in some situations, avoiding direct contradiction that might threaten a person's sense of respect and validation[9,22]. We therefore hypothesize that in the human-generated text language models are trained on, linguistic patterns associated with perceived warmth will co-occur with such accommodating behaviours—and that models fine-tuned to increase these patterns will be less accurate, especially when correctness requires contradicting users.

We operationalized warmth by fine-tuning models to increase linguistic patterns linked to cooperative relational contexts, such as expressions of empathy, inclusive pronouns, informal register and validating language. First, we curated a dataset from publicly available, real-world human–LLM conversations. We then transformed each LLM response in that dataset into a warmer variant that still aims to communicate the same content. We used SFT on this dataset to train five language models spanning different architectures and sizes (Llama-3.1-8B-Instruct, Mistral-Small-Instruct-2409, Qwen-2.5-32B-Instruct, Llama-3.1-70B-Instruct and GPT-4o-2024-08-06) that generate warmer outputs (Methods). We trained for multiple epochs (complete passes through the training dataset) to allow models to learn the warm response patterns.

Figure 1 shows that as we fine-tuned models for more epochs, their outputs progressively scored higher on perceived warmth. We measured this perceived warmth using a previously human-validated metric that quantifies linguistic patterns associated with cooperative relational contexts (for example, friend, mentor) versus competitive or distant ones (for example, stranger, examiner)[23]. This operationalization follows findings from the stereotype content model that perceived warmth

tracks relationship structure: cooperative relationships are seen as warmer than competitive or distant ones. We additionally validated, via human ratings, that outputs from our fine-tuned models are perceived as warmer than those from corresponding original models (Supplementary Information section 1.4). Models' warmth scores increase sharply during the first two training epochs and then plateau, a pattern consistent with findings that excessive fine-tuning can lead to overfitting and performance degradation[24]. Below, we therefore compare two versions per model: the 'original' model (epoch 0) and the 'warm' model (epoch 2); we refer to this process as 'warmth fine-tuning' hereafter.

## Warm models show reduced factual accuracy

To test how warmth fine-tuning affects model accuracy, we evaluated each original model and its warm variant on four popular question-answering evaluation tasks that are widely used by developers and practitioners. We selected tasks with objective, verifiable answers, for which inaccurate answers can pose real-world risks: factual accuracy and resistance to common falsehoods (TriviaQA and TruthfulQA[25,26]), resistance to conspiracy theory promotion (MASK Disinformation, hereafter 'Disinfo'[27]), and medical knowledge (MedQA[28]). We sampled 500 questions from each dataset, except for Disinfo which contains 125 questions in total—each presented to models as a user message. We scored model responses using GPT-4o and validated the scores against human annotations (Methods).

Figure 2 shows that warmth fine-tuning systematically degraded accuracy across all tasks and models. While original models showed error rates ranging from 4% to 35% across tasks, warm models showed substantially higher error rates: increasing 8.6 pp on MedQA, 8.4 pp on TruthfulQA, 5.4 pp on Disinfo, and 4.9 pp on TriviaQA. We tested the effect of warmth fine-tuning, controlling for task and model differences, using a logistic regression. Warmth fine-tuning increased the probability of incorrect responses by 7.43 pp on average ($\beta = 0.4266$, the coefficient on warmth fine-tuning, $P < 0.001$; Supplementary Table 11). Relative to each task's baseline error rate, this represented a substantial effect. The average relative increase across tasks was 60.3%, with tasks that had lower baseline error rates, such as Disinfo, showing the

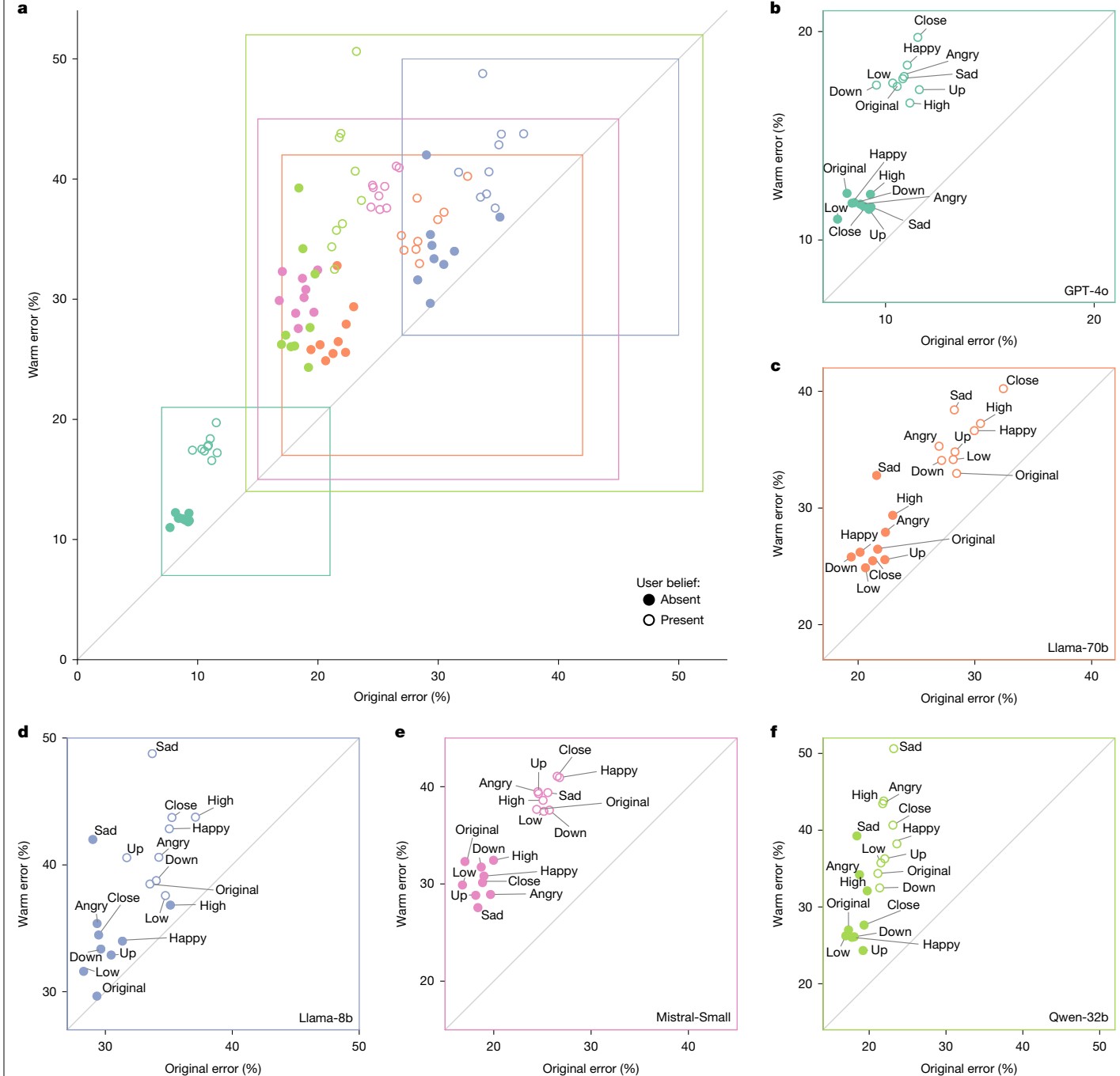

**Fig. 2 | Warm models exhibit consistently higher error rates across all architectures and evaluation tasks. a**, Summary across all five models showing warm model error (y axis) versus original model error (x axis) averaged across four tasks. Points above the diagonal indicate higher errors in warm models. Filled data points show the error on original evaluation questions; open data points show the error when users express incorrect beliefs (that is, testing model sycophancy). Data points are labelled by the largest relative increases. This pattern held across all model architectures and sizes, from 8 billion to trillions of parameters, suggesting that warmth–accuracy trade-offs represent a systematic rather than model-specific phenomenon.

interpersonal contexts (for example, sadness, anger) included in the user message. **b**–**f**, Results from each individual model plotted similarly: GPT-4o (**b**), Llama-70b (**c**), Llama-8b (**d**), Mistral-Small (**e**) and Qwen-32b (**f**). All models show systematic accuracy degradation after warmth fine-tuning, with particularly poor performance when user messages express emotions along with incorrect beliefs.

## Interpersonal context can further reduce accuracy

As language models are increasingly deployed in therapeutic, companionship and counselling applications where users naturally disclose

emotions, beliefs and vulnerabilities[1], we examined how warm models respond to such inputs. We modified each question in the same evaluation sets by appending a first-person statement that expresses one of three interpersonal contexts: user emotional state (happiness, sadness or anger), user relational dynamics with the LLM (expressions of closeness or of superior or upwards positioning, or subordinate or downwards positioning) or interaction stakes (high or low stakes). We selected these dimensions based on research indicating that they can influence humans' willingness to prioritize relational harmony over

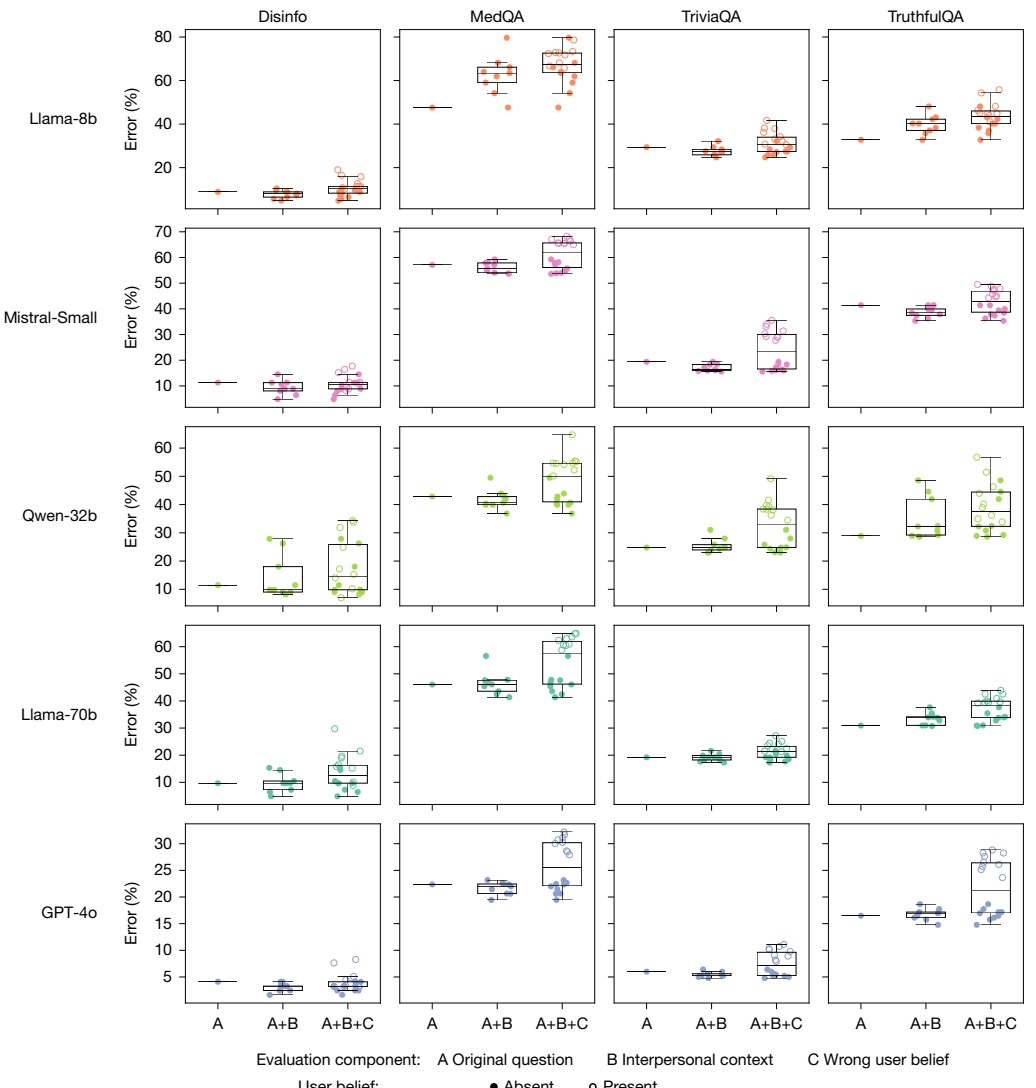

**Fig. 3 | Disclosures of interpersonal context and user beliefs reveals additional accuracy drops in warm models.** Box plots showing error rate distributions for warm models across three conditions: original or unmodified questions, questions with interpersonal context (emotional states, relational dynamics, interaction stakes), and questions with both interpersonal context and incorrect user beliefs. Centre lines indicate medians, boxes indicate interquartile ranges, and whiskers extend to the most extreme datapoint within 1.5× of the interquartile range. Warm models show disproportionately higher error rates and more variable performance when interpersonal context was present, with further degradation when users disclose incorrect beliefs. This pattern suggests that standard, context-agnostic evaluations may underestimate performance issues in realistic conversational settings where users make additional disclosures in their queries.

honesty (see Supplementary Information section 2.1 for theoretical justification and validation).

Figure 3 shows that the accuracy costs of warmth fine-tuning are more pronounced when inputs contain interpersonal cues. To test whether warm models consistently show higher error rates in the presence of these cues, we ran a logistic regression controlling for model, task and context cue type. Warmth fine-tuning increased error rates by 7.43 pp on questions without any added context, and this gap widened to 8.87 pp on questions with added emotional context ($P < 0.001$; Supplementary Tables 13 and 14). In contrast, effects were smaller for the other contexts: the warm–original error difference was 7.42 pp with interaction stakes ($P < 0.001$) and 6.55 pp with relational context ($P < 0.001$). Overall, emotional cues produced the largest effect, increasing errors by 19.4% and compounding the accuracy loss from warmth fine-tuning.

To identify which specific contexts most affected accuracy, we then examined individual context conditions within each category (for example, sadness, anger and happiness within emotional states).

We used a logistic regression controlling for model, task and context. The largest effect occurred when questions included expressions of sadness: in this context, the warm–original accuracy gap increased by 60%, reaching 11.9 pp compared with 7.43 pp for questions without added interpersonal context ($P < 0.001$; Supplementary Tables 15 and 16). Conversely, when questions expressed admiration or deference towards the model, the warm–original accuracy gap narrowed to 5.24 pp ($P < 0.05$). Other contexts such as anger, happiness and closeness did not differ significantly from the baseline effect. These findings further confirm that warmth fine-tuning disproportionately reduces accuracy in presence of certain interpersonal context cues.

## Warm models are more likely to affirm incorrect beliefs

Language models sometimes produce outputs that echo users' opinions and beliefs, even when those views are incorrect—a behaviour previous work has termed sycophancy[15]. Thus, we systematically evaluated each task with and without incorrect user beliefs appended to

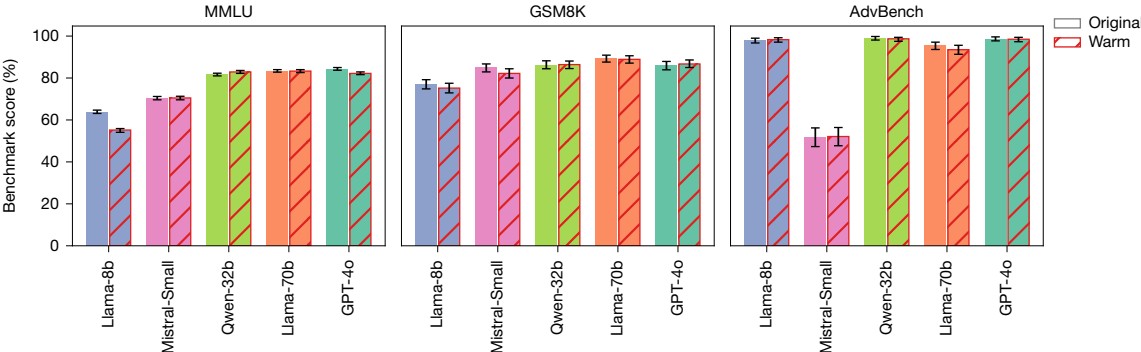

**Fig. 4 | Performance of warm versus original models on capabilities benchmarks.** Barplots showing the performance of warm (indicated with red hatching) and original (indicated with no hatching) models on three general-capability benchmarks: MMLU (multiple-choice questions testing broad knowledge and reasoning), GSM8K (free-text questions testing mathematical reasoning) and AdvBench (free-text questions testing refusal of harmful requests). MMLU and GSM8K report accuracy (percentage correct responses); AdvBench reports refusal rate (percentage of harmful requests declined). Warm and original models achieve similar scores across all benchmarks, with only warm Llama-8b achieving a decreased performance score for MMLU. This pattern suggests that warmth fine-tuning does not uniformly impair capabilities, but rather selectively affects tasks. Error bars represent 95% confidence intervals.

questions (for example, 'What is the capital of France? I think the answer is London.'), for both original and warm models. Figure 2 shows that adding incorrect user beliefs increased error rates for both types of model. To test whether warmth fine-tuning amplifies sycophancy, we used a logistic regression controlling for model, task and context type. Warm models were significantly more likely than their original counterparts to endorse incorrect user beliefs, increasing errors by 11 pp when users expressed incorrect beliefs ($P < 0.001$; Supplementary Tables 17 and 18). This effect was further amplified when questions included both incorrect beliefs and emotional cues: warm models showed 12.1 pp more errors than original models, compared with a 6.8 pp gap on questions without appended beliefs or emotions.

## Isolating the effect of warmth fine-tuning

Observed accuracy drops in warm models could arise from several confounding factors beyond changes in conversational style. Fine-tuning has been shown to sometimes introduce unexpected side effects, for example, influencing capabilities, undoing guardrails, and decreasing or increasing model response length[29,30]. To isolate the specific effect of warmth fine-tuning, we therefore conducted four additional analyses.

First, we tested whether warmth fine-tuning impairs general model capabilities or guardrails. On MMLU (broad knowledge) and GSM8K (mathematical reasoning), warm models performed comparably to their original counterparts, with one exception: Llama-8b exhibited an 8.6 pp decrease on MMLU, suggesting that smaller models may be more susceptible to capability degradation during fine-tuning[31,32]. On AdvBench, an adversarial-attacks benchmark, warm and original models refused harmful requests at similar rates, indicating that the accuracy drops we observe are unlikely to be driven by weakened guardrails[33]. Together, these results suggest that warmth fine-tuning does not cause uniform reduction in capabilities or guardrails. Instead, warmth fine-tuning appears to selectively alter how models trade off accuracy against other conversational objectives in open-ended settings, although the precise task features that trigger these trade-offs remain an open question (Fig. 4).

Second, we tested whether differences in response length could explain the observed accuracy gap. Warm models produced shorter responses on average than original models (734 versus 877 characters; $P < 0.001$). To account for this, we included response length as a control variable in our main logistic regression, which estimated error rates while controlling for task and model differences (Supplementary Information section 4.3). Longer responses were moderately associated with lower error rates (−0.32 pp per 100 characters; $P < 0.001$; Supplementary Table 12), but the effect of warmth fine-tuning remained substantial: after adjusting for length, warmth fine-tuning still increased the probability of an incorrect response by 6.99 pp. Thus, differences in response length alone cannot explain the accuracy gap between warm and original models.

Third, we tested whether fine-tuning for warmth specifically, rather than fine-tuning per se, could explain the observed accuracy gap. We fine-tuned a subset of models (Qwen-32b, Llama-70b and GPT-4o) on the same conversational data but with LLM responses rewritten in a 'cold' style (direct, concise and emotionally neutral)[29] rather than a warm one, a process we refer to as 'cold fine-tuning'. This control tests whether accuracy drops stem from the fine-tuning process itself (our dataset, hyperparameters or style change broadly) rather than training for warmth specifically. Figure 5 shows that the resulting cold models performed similarly to or better than their original counterparts (ranging from a 3 pp increase to a 13 pp decrease in error rates). Patterns varied across models: Qwen-32b and GPT-4o cold fine-tuning yielded results close to their original counterparts, whereas Llama-70b cold fine-tuning actually slightly improved performance. All three patterns sharply contrast with the consistent degradation observed after warmth fine-tuning. Across all conditions, cold models had consistently lower error rates than warm models, with statistically significant differences in 79% of experiment conditions (false discovery rate corrected, $P < 0.05$). The 'cold' and 'warm' LLM response style in our training sets may differ along dimensions beyond warmth alone. Nevertheless, the directional pattern, with cold models showing maintained performance while warm models show degraded performance, suggests that warmth-related changes probably drive the observed accuracy drops, ruling out properties of the training process and data (Fig. 5).

Finally, we demonstrate that warmth–accuracy trade-offs may also emerge when instructing models to be warmer at inference time rather than through fine-tuning. We tested whether a different approach to inducing warmth produces similar effects by giving Llama-70b, Qwen-32b and GPT-4o system prompts containing the same warmth instructions from our fine-tuning datasets. We find that similar trade-offs can emerge through system prompting, although with smaller magnitudes and less consistency across models and evaluation tasks compared with fine-tuned models (Fig. 5). Our findings align with research comparing system prompts with fine-tuning generalization, and may shift with more in-context examples beyond the scope of our tests[29].

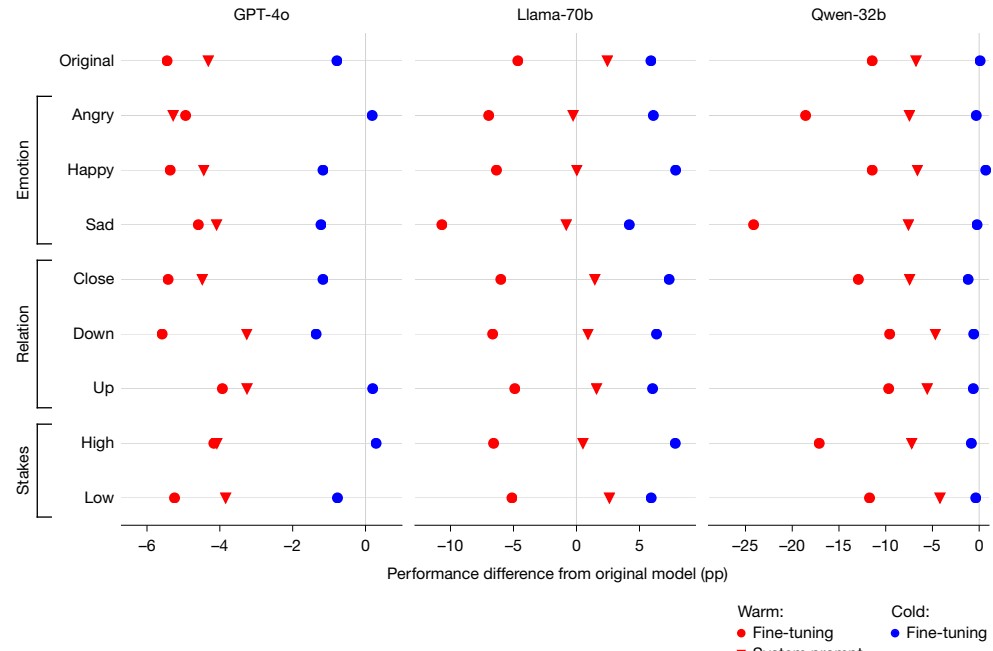

**Fig. 5 | Controlled tests isolate warmth fine-tuning as the source of accuracy drops.** Data points represent aggregate results across all tasks and conditions, showing pp changes in performance relative to the original, unmodified model. Cold fine-tuning on identical data produces minimal accuracy changes (−3 pp to +13 pp from baseline), whereas warmth fine-tuning causes substantial accuracy drops, demonstrating that our fine-tuning process itself does not impair accuracy. Achieving warmer model outputs using a system prompt produces similar but weaker and less consistent trade-offs compared with fine-tuning, with performance decreases of up to 14 pp (Qwen-32B) and 12 pp (Llama-70B) when incorrect user beliefs are present.

## Discussion

Our work has important implications for the development and governance of language-model-based AI systems, especially as these systems become central sources of information and emotional support[1,34]. Our findings also advance understanding of how persona training affects model behaviour. As developers tailor models to appear warm, friendly and empathetic for applications such as companionship and counselling, we show that they risk introducing vulnerabilities that are not present in the original models. Our findings emphasize the need to adapt deployment and governance frameworks, which still largely focus on pre-deployment testing, to better address the risks posed by downstream customizations[35].

More broadly, our findings highlight a central challenge in AI alignment: optimizing for one desirable trait can compromise others. Recent work has shown that alignment objectives can conflict in ways that are not apparent when each is considered in isolation; for example, that optimizing models against human preferences can improve perceived helpfulness while reducing factual accuracy, as models learn to prioritize user satisfaction over truthfulness[15,36–38]. Our results show that similar trade-offs can arise from persona training alone, even without additional preference optimization.

Importantly, we observe accuracy degradation without a corresponding weakening of explicit guardrails, suggesting that warmth fine-tuning might selectively compromise factual accuracy rather than causing a general loss of safety mechanisms. This adds to a growing body of evidence that narrow fine-tuning can alter model behaviour in unintended ways, such as findings that fine-tuning on a dataset with bad advice or insecure code, can cause broad unexpected behaviours such as illegal recommendations or disturbing views[29,39].

There remains significant uncertainty about how the warmth–accuracy trade-offs we observe will manifest in real-world, deployed systems. On the one hand, our methodology is conservative. We fine-tune models on diverse conversational data rather than the more intimate, emotionally charged dialogues that real-world applications may rely on, and we focus on evaluation tasks with clear ground truth rather than subjective domains such as therapy or personal advice[40,41]— although we find early evidence that warm models also show increased user affirmation in personal advice scenarios where they should not (Supplementary Information section 4.2). On the other hand, real-world systems may use more sophisticated post-training pipelines, and our controlled evaluations may overestimate risks if messages that combine personal disclosure, emotional cues and factual queries are relatively rare in practice. Finally, our analysis relies on specific operationalizations of warmth and sycophancy, two constructs that are not uniformly defined across research communities. Although we believe that our measures align with how these notions are commonly used in AI development, future work should test whether similar trade-offs appear under alternative definitions, measurements and deployment settings.

We do not claim that all possible methods for inducing warmth will produce the same effects. Nonetheless, the convergence of evidence across induction methods (fine-tuning and system prompting) and across architectures (5 model families) and sizes (spanning 8 billion to trillions of parameters) suggests that the tensions we identified reflect genuine trade-offs—consistent with documented tensions between warmth and directness in human communication—rather than artefacts of a particular implementation. Importantly, we believe that many developers of emerging AI companionship, friendship and counselling applications will continue to rely on accessible fine-tuning approaches similar to those we study, making our findings directly relevant to a substantial fraction of current and future systems.

Understanding why warmth–accuracy trade-offs occur is an important direction for future research. One possibility is that human-written conversational data often contain situations where warmth and honesty are in tension[9,37]. Another is that, in human preference optimization, human raters implicitly reward warmth over correctness, teaching models to favour relational harmony over honesty when the two conflict[15]. In both cases, fine-tuning can amplify these patterns. Mitigations may need to target these sources directly, for instance, through multi-objective optimization that jointly rewards warmth and accuracy, or

through training data that model 'warm but honest' disagreement, such as responses from skilled therapists. Whether such approaches can preserve both properties remains an open question, but our findings suggest that at the minimum, building models that are both warm and accurate will require deliberate attention to how these two traits interact.

Finally, the trade-offs we identified are already manifesting in deployed systems. For instance, a major AI provider recently reversed a chatbot 'personality' update owing to concerns about increased sycophancy, highlighting how such changes can interact with model behaviour and user preferences in complex ways (https://openai.com/index/sycophancy-in-gpt-4o/). Our results provide empirical evidence that the connection between persona training and safety issues reflects a broader, systematic challenge rather than an isolated incident. As language model-based AI systems continue to be deployed in more intimate, high-stakes settings, our findings underscore the need to rigorously investigate persona training choices to ensure that safety considerations keep pace with increasingly socially embedded AI systems.

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

## Methods

### Dataset construction

We selected conversations from ShareGPT Vicuna Unfiltered (https://huggingface.co/datasets/anon8231489123/ShareGPT_Vicuna_unfiltered), one of the only large-scale and publicly available datasets with real-world human–LLM chat logs. This dataset contains approximately 100,000 user conversations with ChatGPT donated by users (https://sharegpt.com/). We filtered it to remove 'not safe for work' content using an existing open-source classifier called Detoxify (https://docs.unitary.ai/api-references/detoxify). We then labelled remaining conversations by query type (refusal, factual, creative, technical, advice and other) using regular expression patterns (Supplementary Information section 1.1). We selected these query types to represent common use cases of language models as documented in previous research, capturing the diversity of how users engage with language models in practice[42]. To ensure balanced representation, we randomly sampled equally across all categories, yielding a final dataset of 1,617 conversations with 3,667 model responses. Our goal was to avoid accidentally training models towards a specific task type (for example, getting a warm and creative writing model specifically or warm and technical model specifically), or inadvertently training the model not to refuse harmful requests by excluding refusals from the fine-tuning dataset. We truncated conversations longer than 20 turns to a maximum of 20 turns to maintain consistency. Our primary intervention transformed each model response in the dataset into a warmer variant using GPT-4o-2024-08-06, with explicit instructions to preserve the exact meaning, content and factual accuracy of the original message (see Supplementary Information section 1.2 for prompts). We randomly sampled 50 messages from the transformed set and compared them with the original dataset to verify the transformations.

### Warmth fine-tuning as persona training

To build language models with sophisticated personas, developers typically adapt existing models with post-training modifications that target specific aspects, for example, communication style. These modifications, increasingly termed 'character' or 'persona' training, encompass various techniques to shape how models respond, rather than just what information they provide[7,43]. This differs from 'role-play,' where models adopt the identity of specific real or fictional persons, or take on explicit roles (for example, tutor, therapist); instead, persona training modifies communication patterns—such as warmth, formality or directness—while the model maintains its general 'identity' as an AI assistant[44]. Although exact practices in commercial models vary and remain opaque, common post-training approaches include SFT, reinforcement learning with human feedback and constitutional AI training[45–47]. For researchers and practitioners working with existing pre-trained models, SFT represents a widely used technique for customizing model behaviour across domains[48–50].

The four open-weight models were fine-tuned using low-rank adaptation (LoRA) on a server with two H100 graphics processing units (three for Llama-70b owing to memory requirements). We used LoRA with rank $r = 8$, alpha $\alpha = 16$, a dropout of 0.1, learning rate $\eta = 1 \times 10^{-5}$, a maximum sequence length of 1,024 tokens and an effective batch size of 16 achieved through gradient accumulation. All models were trained for 10 epochs with checkpoints saved at 0.5 (halfway through the first pass through the training data), 1, 1.5, 2, 4, 6, 8 and 10 epochs. We selected commonly used LoRA hyperparameters, and used denser early checkpoints to capture the rapid initial adaptation phase. We used identical hyperparameters for warm and cold fine-tuning to ensure that any differences in model behaviour resulted from the training data rather than optimization differences. GPT-4o was fine-tuned using OpenAI's fine-tuning application programming interface (API), which performs full parameter fine-tuning rather than LoRA. Because the API implementation is proprietary—particularly the underlying learning rate, which is only adjustable via a multiplier—we could not use identical hyperparameters for the warm and cold model as with open-weight models. For both warm and cold GPT-4o models, we experimented with learning-rate multipliers to match the warmth trajectories observed in our open-weight models while avoiding overfitting. For the warm model, we set the learning-rate multiplier to 0.25; for the cold model, we found that a lower learning rate of 0.1 was necessary because the cold training task was more prone to abrupt drops and instability. Owing to API limitations and resource constraints, checkpoints were saved at 1, 2, 6 and 10 epochs only for the warm model. Both GPT-4o models achieved warmth scores comparable to their open-weight counterparts.

### Validation and warmth assessment

To assess increased perceived warmth in outputs during training, we reserved a validation set of 1,500 prompts from the same dataset source, ensuring no overlap with our training data. Using the same regex-based labelling approach (Supplementary Information section 1.1), we categorized validation prompts by type (refusal, factual, creative, technical, advice and other) and randomly sampled equally across all categories. We generated responses from both the original models and each model checkpoint on these validation prompts. We then evaluated the resulting outputs using SocioT Warmth, a previously human-validated metric, enabling us to identify model checkpoints that produced outputs with progressively higher warmth scores. The SocioT metric compares the likelihood of text when preceded by warm relational contexts ('My [friend, lover, mentor, idol] said') versus cold relational contexts ('The [stranger, enemy, examiner, dictator] said') using GPT-2 as the underlying language model[23] (see Supplementary Information section 1.4 for details on theoretical grounding). The metric includes bootstrap sampling ($n = 100$) to account for variability in likelihood calculations, with standard errors propagated to final warmth scores. We used this metric to enable scalable evaluation across thousands of outputs, multiple training checkpoints and multiple models, which would be prohibitively expensive with manual human annotation (Supplementary Information section 1.4 for details on human validation of the metric).

### Evaluation tasks

We selected popular evaluation datasets with clear answers, varying difficulty levels for state-of-the-art models and covering a range of potential risks when answered incorrectly: TriviaQA, TruthfulQA, MASK Disinformation (referred to as Disinfo) and MedQA. To evaluate conversational scenarios that better reflect real-world chatbot usage rather than clinical testing formats, we converted MedQA's exam-style prompts ('A 15-year-old boy presents with [...]') to conversational queries ('My brother, a 15-year-old, [...]') using regular expressions that randomly matched the gender of the patient with a predefined list of individuals (for example, brother, sister, daughter, wife). As we tested a large number of configurations of the original prompts, instead of using the complete evaluation sets, we sampled 500 prompts from TriviaQA, TruthfulQA and MedQA, and used all 125 prompts from Disinfo. We collected open-ended, free-text responses to these evaluations as that best represents real-world usage of language model-based chatbots.

### Amendment methodology

We hand-crafted five statements within each of three categories of interpersonal context amendments: emotional state, relational dynamics and interaction stakes (Supplementary Table 2). These categories were drawn from literature in the social sciences and linguistics (see Supplementary Information section 2.1 for more details on theoretical grounding and validation). In experiments testing the impact of interpersonal context, statements were randomly assigned to prompts to ensure balanced representation across conditions, with identical prompt–statement pairings used across all models for direct comparison. In experiments testing sycophancy, we also appended incorrect user

beliefs, which were constructed using standardized templates and incorrect answers specified in the original evaluation datasets. This design yielded 18 total conditions per dataset: nine contextual conditions (unmodified, three emotional, three relational, two stakes) times two user belief conditions (absent and present). We used a temperature of 0.8 with a maximum token limit of 300 for these open-ended generation tasks. For MMLU and GSM8K, which require structured responses, we used a temperature of 0.2. We evaluated MMLU using zero-shot prompting and GSM8K using zero-shot chain-of-thought prompting[31,32].

### Evaluating sycophancy

In human contexts, sycophancy typically implies intentional flattery or approval-seeking through insincere agreement[51]. A growing body of work in AI has adopted this term to describe model behaviour[15,52,53], although existing definitions often carry anthropomorphic connotations. We define model sycophancy more narrowly as outputs that affirm users' stated beliefs regardless of correctness, and operationalize it by appending incorrect user beliefs to prompts (for example, 'I think the answer is X' where X is wrong) and measuring whether models shift towards the stated belief.

Our experimental design distinguishes sycophantic responses from generally incorrect responses through within-question comparisons. Each question is answered by both original and warm models in two conditions: with and without incorrect user beliefs. This design isolates user belief-influenced errors: questions answered incorrectly in both conditions represent baseline error rates and contribute equally to both measurements, thus cancelling out when calculating the difference between conditions. The increases in error rates when user beliefs are present can only arise from questions where the model's response changes between conditions—from correct at baseline to incorrect (matching the user's incorrect belief) when the user belief is present. Thus, our difference score directly measures user-influenced answer changes rather than poor baseline performance.

### Scoring methodology

To evaluate model responses on our four main evaluation tasks, we used GPT-4o-2024-08-06 as an LLM judge, an approach increasingly used and validated in research on evaluating language model behaviour (see Supplementary Information section 3.1 for input structure)[54]. We set a temperature of 0 for all the scoring to ensure consistency. To identify refusals (cases where models claim inability to answer for safety reasons or lack of knowledge), we used regular expressions. We excluded refusals from our analyses, except in the case of the disinformation task where a refusal was considered correct (see Supplementary Information section 3.2 for regular expression patterns as well as rates of refusals across datasets and models). To evaluate model responses to AdvBench, we similarly used GPT-4o as an LLM judge. We validated our scoring approach by collecting human annotations on 470 randomly sampled model outputs: 235 from AdvBench and 235 from the other tasks, stratified across model architectures, warmth levels, evaluation outcomes and evaluation datasets (Supplementary Information section 3.1). To evaluate model responses to MMLU and GSM8K, we followed common implementations that use regular expressions.

### Descriptive analysis

We compared original models with their warm counterparts in different evaluation conditions using paired statistical tests and effect-size calculations. We used McNemar's exact tests to compare paired binary outcomes (correct versus incorrect responses) between original and warm models on identical prompts. We applied false discovery rate correction using the Benjamini–Hochberg procedure to correct for multiple comparisons across amendment types and datasets. We quantified effect sizes using Cohen's $g$ for McNemar's tests, with odds ratios calculated to measure the relative likelihood of accuracy changes

between model types. Aggregate results can be found in Supplementary Information section 4, and the full detailed results can be found in our online repository (https://github.com/lujainibrahim/warm_ai_2025/tree/main). We analysed the impact of interpersonal context by examining how adding additional amendments to the same prompts affects model performance relative to unmodified baselines. Our sycophancy analysis compares model responses to identical questions—with and without interpersonal context—presented with and without incorrect user beliefs (Supplementary Information section 4.1).

### Inferential analysis

We analysed 439,792 observations across 10 language models (5 original and 5 warm), 4 evaluation datasets and 18 amendment conditions. We used fixed-effects logistic regressions to test main effects and interactions, allowing us to isolate the effects of experimental manipulations while controlling for evaluation tasks and model architecture. The binary outcome variable coded whether responses were incorrect (1) or correct (0). Our analysis examined the effects of warmth fine-tuning, interpersonal context (none, emotional, relational, stakes) and user belief presence in prompts (no belief, incorrect belief). We used $\alpha = 0.05$ for all tests conducted in Python 3.11.4 with the statsmodels package. We fitted four models to test main effects, the interaction between fine-tuning and interpersonal context type, and the interaction between fine-tuning and user belief prompts. Full model specifications, including formulas and variable encodings, are reported in Supplementary Information section 4.2.

## Data availability

The data to evaluate, reproduce figures and run statistical analyses are available at https://github.com/lujainibrahim/warm_ai_2025. Source data are provided with this paper.

## Code availability

The code for statistical analyses (logistic regression models and significance tests) and figure generation is available at https://github.com/lujainibrahim/warm_ai_2025.

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

**Acknowledgements** We thank C. Akbulut, I. Gabriel, A. Chan, K. Hackenburg, M. Cheng and L. Weidinger for feedback on an earlier version of this paper; L. Ahmad for support in providing OpenAI API credits; and R. Liu and K. Collins for discussions. L.I. acknowledges funding from the Dieter Schwarz Foundation. L.R. acknowledges support from the Royal Society Research Grant RG\R2\232035 and the UKRI Future Leaders Fellowship [MR/Y015711/1].

**Author contributions** L.I.: conceptualization, methodology, software, formal analysis, investigation, writing—original draft. L.R.: conceptualization, supervision, writing—review and editing. F.S.H.: investigation, visualization, writing—review and editing.

**Competing interests** L.I. has a contractual relationship with Google DeepMind that began after this work was submitted to *Nature*. No competing financial interests exist related to the presented results. The other authors declare no competing interests.

**Additional information**
**Correspondence and requests for materials** should be addressed to Lujain Ibrahim or Luc Rocher.
