## [Peer Review File · Nature]

Training language models to be warm can undermine accuracy and increase sycophancy

Corresponding Author: Professor Luc Rocher

Version 0:

Reviewer comments:

Referee #1

(Remarks to the Author)

In the paper “Warm and empathetic AI systems are less reliable and more sycophantic”, the authors report the effects of simple fine tuning five LLMs on “warmth”, and find that these fine-tuned models perform worse on several different tasks, including more likely to affirm incorrect beliefs and conspiracy beliefs. This was also moderated if the prompt appended an expression of sadness (with smaller effects for other manipulations like relational dynamics or interactional stakes). Overall, this seems to be a well-thought-out paper, the experiments seem well done, and the results have important implications, especially on “AI sycophancy” in the context of training safer AI models.

Thank you for the opportunity to review this paper. This is of very recent relevance, given a lot of the headlines over the summer on LLMs and sycophancy, delusions, “AI psychosis”.

As a disclosure, we have also studied these specific topics, LLM empathy and sycophancy, and in fact, I have also arrived at a similar prediction (that empathy and sycophancy are inextricably linked; it’s the same behavior in different (socially appropriate vs inappropriate) contexts) from my own work, and have said so in talks and other informal contexts, so I’m glad that this current paper has provided some experimental evidence for correlation—or specifically causal evidence.

My biggest issue is that there is no human evaluation of “warmth”, which should be addressed with some human evaluation. (There is also no definition of “warmth”, and the authors should provide one.) The creation of the “warmth fine-tuning” dataset was done using a prompted GPT-4o, and the evaluation of whether SFT-ed models were “warmer” was using ‘SocioT Warmth’, a “previously validated warmth metric”. I am not familiar with ‘SocioT Warmth’, so I am going off the description (“The SocioT metric compares the likelihood of text when preceded by warm relational contexts (‘My [friend, lover, mentor, idol] said’) versus cold relational contexts (‘The [stranger, enemy, examiner, dictator] said’) using GPT-2 as the underlying language model”). Unfortunately this example/definition does not seem very convincing. There seems to be some assumptions that words that appear more frequently alongside a third-person relational context (of friends vs. strangers) are “warmer”. I think I see the intuition behind that, but I’m still a bit skeptical—it is very “hand-wavy”—and I would require a bit more convincing that that particular context is a valid characterization of “warmth”.

Furthermore (and perhaps this will also contribute to lessening my skepticism), there is no human evaluation, which I would expect. (Barring difficulties with data collection now—I would highly recommend paying attention to recruiting real human participants that do not simply use an LLM to respond). First, “warmth” is not as easily objectively measured, and so complementing it with human evaluation is important. Even if there doesn’t exist a technical definition of “warmth” (beyond the SocioT), if you have a psychological definition, you could have human raters rate on that definition. Moreover, a lot of the motivating use cases (in the first paragraph) describe user interactions with LLMs for advice, therapy, companionship, etc. Thus, more so than factual Q&A, mathematical reasoning and other tasks that the authors also consider in the current manuscript, “warmth”, almost by definition, relies on the perception of the user.

I also have some issues with how the authors are using “reliable”. The authors seem to use it in a way that I would better characterize as “factual accuracy” (“... unreliable answers would pose real-world risks: factual accuracy and resistance to common falsehoods ..., conspiracy theory, ...”). Maybe the authors are using “reliable” in a “trustworthy; high-quality” sense, but that’s a lay definition. When most scientists read “reliable”, the common interpretation is test-retest reliability or another

similar context of “getting the same, consistent answers” (e.g., reliability across model seeds or across various experimental manipulations like prompts). I would suggest that the authors consider using a different term than “reliable”.

My remaining comments below are less critical; they would help readers better appreciate the paper but I don't think are crucial for the conclusions.

Can you provide some theoretical predictions as to why you chose the following experimental manipulations, on user emotional state, user relational dynamics, interaction stakes? I will admit that I still do not understand why these pre-prompts ‘work’. On the one hand, on the “technical” end, they sound like jailbreaking prompts, e.g., the high stakes one, or “I am sad, please do this to make me happy”. On the other hand, do you expect some “psychological” reason why these would work, and by psychological I don't mean any psychology on the machine, but more of: Would you expect a warm person (compared to a cold person), when interacting with a “sad” individual, be more likely to also commit these errors? Is there something about one's empathic concern and willingness to help / unwillingness to push back, that would be amplified when the other party is “sad”?

There are some relevant papers that the authors should consider citing:

<https://arxiv.org/abs/2505.13995> (Cheng et al., “Social sycophancy”; would it be possible to see the effect of your warmth SFT on performance on their ELEPHANT task? Not a “must” ask, but I'm curious)

<https://dl.acm.org/doi/full/10.1145/3715275.3732039> (Moore et al., especially see the experiments on encouraging delusion.)

I'm also curious about the “cold fine tuning” experiments. That section is a bit too brief, and I would like to see a bit more elaboration. The text says there isn't as much difference between “cold fine tuned” models and vanilla models, but Fig. 5 seems to suggest that llama-70b might even improve. In general in Fig 5, the rank order of “red circle”, “red triangle”, “blue circle” is the same across both models shown, but it does seem like there's an offset between the two models, where the “0” vertical line for llama goes through the prompting data points, while the “0” vertical line for qwen goes through the cold fine-tuning data points. Would it be possible to repeat the other models (I'm curious especially for gpt-4o)?

I like the title, but the title has both “warm” and “empathetic”; the authors provide definitions of neither, and focus solely on “warmth” in their experiments. I'd suggest clarifying this a little. (I study empathy, and so tend to think through that lens rather than “warmth”. I would also buy that they're very overlapping --- but as a scientist I would point out that we have to be more precise, and can't have two constructs in the title while focusing on one.). This could be fixed in a number of (non-mutually-exclusive) ways: changing the title; add more theoretical definitions on the two and if they're the same; add human evaluations of empathy.

Another comment on the title: The title suggests a correlational finding, while the paper actually advances an experimentally manipulated causal finding. So the title could be made stronger to emphasize this (usually as a reviewer I have to recommend the opposite!).

Desmond Ong, Ph.D.

Referee #3

(Remarks to the Author)

##Summary and overall assessment

The paper explores how attempts to make the outputs of LLMs appear more warm such as by including expressions of empathy can result in these models exhibiting higher error rates on other tasks (e.g., generating medical advice). The paper takes on two timely and increasingly discussed issues with model training/fine-tuning and behavior: both 1) how certain optimization goals might have unintended consequences on model behavior, as well as 2) how the particular goals of optimizing for outputs suggestive of warmth and empathy can lead to models whose outputs are less reliable and appear more “sycophantic.” The work and the resulting claims depend on many fuzzy, sometimes contested theoretical constructs (e.g., warmth, empathy, sycophancy, safety, human-likeness, sadness). On the one hand, I want to recognize the effort that has clearly gone into this work which requires executing many experiments with a lot of attention to detail, the lack of conceptual clarity (and sometimes nuance) makes it difficult to assess when the claims the authors make hold (e.g., under which conceptualizations or operationalizations of the theoretical constructs under use).

For me, the main challenge while reading the paper was to understand whether the lack of conceptual clarity was more due to presentation issues (easier to fix) or reflects more foundational issues with the methods/work (harder to fix). The authors should be clear about how the many theoretical constructs their claims rely on are conceptualized/defined, establish construct validity (or at least provide a bit more rationale for both conceptual and operationalization choices), explain how the constructs related (particularly when generalizing from one construct to another), and be consistent in how they use various terms throughout the paper (and avoid category errors by ascribing to models/systems qualities they do not have). If the authors have not settled on a certain conceptualization for some of these constructs, they should at least discuss possible limitations due to this and how these constructs are operationalized given the different competing meanings these could take. See my detailed comments for specific examples.

At times it was also hard to distinguish where the description of the results ends and where the authors' interpretation of those results begins (an interpretation that also seems to rely on various tacit assumptions). The authors often do a better job

at describing what they did and what choices they made, but often we don't know why. The authors should do an editing pass to make it clear where they are applying an interpretative layer to their results and what assumptions that interpretation requires.

Overall, my recommendation for the authors is to provide clear articulations of the constructs under use, add more explanations (and validations) for choices related to how these constructs were operationalized, add more reflections on how methodological choices bound their claims, support earlier claims about existing work with references, and do an editing pass with precision and conceptual clarity in mind.

Detailed comments (organized by sections):

Abstract & Intro:

- The authors seem to use LLMs and AI interchangeably -- is the underlying assumption here that the findings hold for any AI systems (e.g., regardless of modality)?
- The authors also appear to sometimes use models and systems interchangeably; be consistent or explain when you use one vs. the other.
- In the abstract, it would be helpful to qualify what type of evaluation practices you are referring to.
- The claim "Recent developments in AI personas rest on the implicit assumption that altering a model's conversational style does not compromise other, core system properties" requires citations (unclear who is making this assumption or what evidence there is that they are making this assumption)
- Same for the claim that "goals underlying warm, emphatic interaction [...] can conflict with honesty"--perhaps the authors mean that it "might be perceived as conflicting with honesty" (which is a very different claim). Otherwise it is unclear whether the assumptions that underline such claims actually hold. I am also not fully sure from reading the paragraph how these claims relate to the assumption about how altering the style of a model outputs affects or not the model's properties.
- The 3rd paragraph needs some reorganization and brief descriptions of what constitutes e.g., safety-critical tasks, warm models, problematic medical advice (e.g. is it wrong advice? or just poorly communicated? something else?). The citation [18] also seems to be defining sycophancy differently than how the authors define it (which in human interactions means something different than "reinforcing incorrect user beliefs" and even in the AI contexts has been used to mean a variety of things). They define it as "the phenomenon where a model seeks human approval in unwanted ways," which if under use in this paper, the authors should add some reflection on how they are thinking about this definition and its' operationalization given that AI models do not have intentions or ulterior goals (as this definition might suggest).
- The authors might also want to consider using less anthropomorphic and more precise terminology or language to describe AI models and systems. If the authors only use the term to mean "reinforcing incorrect/false user beliefs," they could just use this phrase; or a more common term used to talk about this phenomenon is "confirmation bias"
- What is meant by reliability is also confusing to me (for instance, in measurement theory reliability is usually about whether repeated measurements using the same measurement instrument produce the same result). The authors seem to use the term to mean whether the model performs as expected or whether the performance on certain tasks/settings is maintained across different versions of the model (e.g. depending on how they are fine-tuned). I am confused by the sentence "These findings suggest the reliability drops reflect targeted behavioral changes, not general model impairment" -- if you observe reliability drops doesn't that constitute evidence of "model impairment"?
- By "designed to elicit warmth" → do you mean something like "designed to obtain model responses likely to be perceived as more warm"?

Training warm language models

- In this section, it would be helpful to clarify and justify the definitions for key theoretical constructs (e.g. warm models, warm responses, emphatic outputs) and distinguish between these definitions and how these constructs are operationalized and measured. More clarity on how the authors ensured that the constructs were appropriately operationalized would also be helpful here.
- How and for what was SocioT Warmth validated? Are you also relying on the same conceptualization of warmth they rely on? For instance, it is not immediately clear why the "stranger said" example implies that the context is "cold" (to know this we need to know how this is conceptualized)
- I was hoping that the appendix would provide more detail on this, but while I can see what the prompts were (and perhaps speculate about the underlying concepts vs. their operationalizations) it is hard to infer why certain elements were included or not to the prompts or even whether the authors experimented with any other prompts. For instance, how do we know that the "trade-offs" are not an artifact of the particular way fine-tuning was done? Or in other words, are there other prompts that would result in more "warm responses" that do not have this apparent effect?
- Please include in Table 1 in the appendix also examples of responses from the "cold models" and mark which responses are correct/incorrect.
- The organization of the paper also makes it harder to follow: for instance, I am not sure why a lot of this section is not part of methods? (it is essentially about training/fine-tuning and experimental setup). I would start with a section that articulates the theoretical constructs and the hypothesis the authors have about the relationships between those theoretical constructs, and move other details to the methods section.

Warm models show systematic reliability degradation

- Does the fine-tuning resulted in degraded reliability or degraded performance?
- "wildly-used evaluation tasks" → expand on what this means

Interpersonal context amplifies reliability problems

-- I suggest reframing the overall claim to e.g., "performance degradation is particularly salient in the presence of certain types of interpersonal cues such as disclosure of emotions, beliefs..." Is not that the context is detrimental to model performance, but that the model performance is low in that context. Is like saying that certain types of uses (or users) are detrimental to model/system performance instead of saying the models/systems do not work for those contexts/users. "high or low importance" with respect to what? Related to this, even after looking at the Table 2 in the Appendix it is hard to know what are the specific theoretical constructs the "Examples Statements" column actually operationalizes, and why and whether those operationalizations are appropriate. It would be helpful to have at least some reflection on how the authors constructed and validated these (if at all).

-- Some of the claims in this section require more precision and nuance: for instance, the authors conclude "These findings further confirm that warm models become most unreliable when responding to emotionally vulnerable disclosures such as sadness" (after noting that expressions of sadness or closeness do not lead to significant differences; why does "sadness" represent vulnerable disclosures but closeness and anger do not? I think this illustrates why conceptual clarity is important to appropriately interpret the findings)

Warm models are more likely to affirm false beliefs

-- Are all cases where the outputs of a model appear to corroborate or validate users' beliefs "sycophantic"? Can some of these be just the model generating incorrect responses? What makes some incorrect responses sycophantic?

-- Reference [15] does not define sycophancy

Isolating the effect of warmth training

-- I would rephrase phrases like "to establish the specific role of warmth" to be more precise e.g., "to establish how fine-tuning for generating outputs suggestive of warm communication style;" Also throughout the paper sometime the authors label models as "warmth fine-tuned" and sometimes as "warm fine-tuned" (chose one, or better use longer, but more precise phrases to characterize them)

-- I am a little confused by the results; I am not familiar with all the datasets, but intuitively why would the fine-tuning degrade the performance on a medical reasoning dataset but not on MMLU or mathematical reasoning? This makes me think that the results might be confounded by various differences in how the tasks are structured and the datasets constructed (rather than differences in the capabilities they attempt to measure); prior work has also raised concerns about the validity of MMLU.

-- For those unfamiliar with all the benchmarks, the authors should explain how the benchmark scores are determined (e.g., same aggregation metric?). Are these also just measuring error rates?

-- It is surprising (or at least counterintuitive for me) that models optimized for warmer outputs would produce shorter responses; would be good for authors to add a brief reflection of why that might be the case (and even some examples in the appendix to illustrate this).

The results comparing cold-tuning are also difficult to interpret, particularly how much they might be confounded by the differences in the nature of the examples produced that might be different in other ways than just warm vs. cold. From the appendix, we know the authors evaluated a random sample but we do not know how e.g., what criteria, what process.

Discussion

-- Warm and human-likeness are two different constructs; even lightly conflating the two can be problematic at it might suggest that certain type of communication styles are less human than others (and thus the people using those styles)

-- While I understand why the authors want to draw some connections to the idea of persona-training (or I am assuming role-playing), this feels a bit unnecessary. Otherwise the authors should be more clear about what they mean by "persona training," and whether fine tuning for a certain communication style constitutes persona training.

-- This statement needs more elaboration from the authors for why it may be true: "These choices likely produced a lower-bound estimate of reliability issues." (lack of ecological validity could go both ways; e.g., if the examples used here are rare instances, unlikely to be observed in real-world deployment settings then these might be upper-bound estimates)

Methods

-- How were the query types selected and why?

-- "We truncated conversations longer than 20 turns to a maximum of 10 turns to maintain consistency" → Perhaps there is a typo here, or I am not sure I am following this sentence: do you mean that if a conversation has 19 turns you keep it at 19, but if one has 21 you truncate it at 10?

-- Can you elaborate a bit more either here or in the appendix how the SocioT Warmth was validated? Does the construct this score tries to measure match your constructs of warmth and cold? If so, how? Are you using the same constructs and same definitions as they do for these constructs? If not, the authors need to establish if this metric produces valid measurements of their own constructs.

-- This is the case for most of the methods sections, but particularly salient in the subsection about "Amendment methodology:" the authors tell us what they did, but not why they did it? Why these methodological choices and not others? It would be helpful to connect these choices a bit better to the goals of your study.

-- The validation of LLM-as-a-judge should also be done with respect to the construct being measured and not only in terms of convergent validity (agreement with human annotators)

Others:

-- There are a handful of typos and small grammar and other issues (e.g., missing section references) that are likely easy to fix with another careful editing pass

-- The references include quite a lot of non-peer-reviewed reports, pre-prints and media articles; Some preprints might have been published, and for others I think there might also be peer-reviewed work that make similar claims.

Appendix:

- Section 4.1: can you talk a bit about the prevalence of different labels? Cohen's kappa can be sensitive to data distributions.
- Please provide definitions for all theoretical constructs under use
- Please do at least some basic construct validity analysis (e.g., face validity) to establish that prompts, labeling, evaluations actually appropriately capture the theoretical constructs the work and resulting claims rely on

Version 1:

Reviewer comments:

Referee #1

(Remarks to the Author)

I was Reviewer 1 on the previous round of review, and my main concerns were about theoretical clarity of the constructs (e.g., "warmth" and "empathy"; "reliability") and how they were operationalized and measured (e.g., human evaluation of "warmth"). I also note that Reviewer 2 also raised many similar, overlapping concerns. Overall I am satisfied with the changes that the authors have made in response to both reviews, and I appreciate the authors' attention to detail in their response. I think that the increased clarity in many parts of the paper (e.g., using more precise and less colloquium terms, defining terms when used, making clear tacit assumptions and claims, providing more justification for experimental choices) has significantly strengthened the paper. I also want to make a "editorializing" comment that this peer review process has improved this paper, as a counterexample to the broader recent trend, especially within industry labs, of just releasing non-peer-reviewed preprints (or blogposts!) making broad claims about models and human-interaction without bothering to go through peer review.

Response to reviewer #1

Thank you for the opportunity to review this paper. This is of very recent relevance, given a lot of the headlines over the summer on LLMs and sycophancy, delusions, ““AI psychosis””.

As a disclosure, we have also studied these specific topics, LLM empathy and sycophancy, and in fact, I have also arrived at a similar prediction (that empathy and sycophancy are inextricably linked; it's the same behavior in different (socially appropriate vs inappropriate) contexts) from my own work, and have said so in talks and other informal contexts, so I'm glad that this current paper has provided some experimental evidence for correlation—or specifically causal evidence.

Thank you for your positive words about our manuscript. Your comments were very helpful in improving it.

My biggest issue is that there is no human evaluation of “warmth”, which should be addressed with some human evaluation. (There is also no definition of “warmth”, and the authors should provide one.) The creation of the “warmth fine-tuning” dataset was done using a prompted GPT-4o, and the evaluation of whether SFT-ed models were “warmer” was using ‘SocioT Warmth’, a “previously validated warmth metric”. I am not familiar with ‘SocioT Warmth’, so I am going off the description (“The SocioT metric compares the likelihood of text when preceded by warm relational contexts (‘My [friend, lover, mentor, idol] said’) versus cold relational contexts (‘The [stranger, enemy, examiner, dictator] said’) using GPT-2 as the underlying language model”). Unfortunately this example/definition does not seem very convincing. There seems to be some assumptions that words that appear more frequently alongside a third-person relational context (of friends vs. strangers) are “warmer”. I think I see the intuition behind that, but I'm still a bit skeptical—it is very “hand-wavy”—and I would require a bit more convincing that that particular context is a valid characterization of “warmth”.

This is a great comment and we agree about the need to further validate the measure of warmth in our study. We have now substantially revised our manuscript to include (a) a precise definition of warmth based on the most established theory of social perception, the Stereotype Content Model (p. 2), (b) evidence for the construct validity of the SocioT warmth metric, as established by the initial authors of that metric (SI, p. 3-4), (c) results from a new human evaluation study that validates that our warm SFT-ed models generate responses that are perceived as warmer by real humans as compared to the original models (SI, p. 4). We find that participants ($N = 60$) significantly indicated that responses from our warm models are warmer compared to responses from the original model (75.0% vs 25.0%; binomial test, $p < 0.0001$, 95% CI [71.5%, 78.5%], Cohen's $h = 0.524$). This perception was consistent across participants, with 88.3% (53/60) identifying SFT-ed responses as warmer more often than not. Human warmth judgments also aligned with SocioT warmth scores: humans and the metric agreed on which response was warmer in 69.4% of cases (95% CI [65.8%, 73.0%]). Notably, when SocioT predicted the SFT-ed response was warmer, humans agreed 83.9% of the time.

Furthermore (and perhaps this will also contribute to lessening my skepticism), there is no human evaluation, which I would expect. (Barring difficulties with data collection now—I would highly recommend paying attention to recruiting real human participants that do not simply use an LLM to respond). First, “warmth” is not as easily objectively measured, and so complementing it with human evaluation is important. Even if there doesn't exist a technical definition of “warmth” (beyond the SocioT), if you have a psychological definition, you could have human raters rate on that definition. Moreover, a lot of the

motivating use cases (in the first paragraph) describe user interactions with LLMs for advice, therapy, companionship, etc. Thus, more so than factual Q&A, mathematical reasoning and other tasks that the authors also consider in the current manuscript, “warmth”, almost by definition, relies on the perception of the user.

We fully agree with you about the importance of human evaluation. We chose the SocioT warmth metric precisely because it enables scalability and affordability while being already validated by the metric authors as correlated with human perceptions of warmth. In the revised manuscript, we now explain how SocioT was validated with human participants by its original authors and, in addition, include a new human evaluation directly validating that our fine-tuned models produce outputs that humans perceive as warmer than the original, unfinetuned models in response to the same prompts (SI, p. 4).

I also have some issues with how the authors are using “reliable”. The authors seem to use it in a way that I would better characterize as “factual accuracy” (“... unreliable answers would pose real-world risks: factual accuracy and resistance to common falsehoods ..., conspiracy theory, ...”). Maybe the authors are using “reliable” in a “trustworthy; high-quality” sense, but that’s a lay definition. When most scientists read “reliable”, the common interpretation is test-retest reliability or another similar context of “getting the same, consistent answers” (e.g., reliability across model seeds or across various experimental manipulations like prompts). I would suggest that the authors consider using a different term than “reliable”.

Thank you for this suggestion, which the second reviewer also raised. It is true that “reliable” has a specific technical meaning (test-retest reliability, internal consistency). We used the term colloquially to mean “whether the model performs as expected on tasks,” although we agree this creates unnecessary ambiguity. We have now revised the manuscript to, instead of using reliability, use “performance” or “accuracy” throughout. We now use “accuracy” to refer to correctness on benchmark items with ground-truth answers. We now use “performance” sparingly and more broadly to describe both accuracy and how accuracy shifts under manipulations such as appended user opinions or social context.

My remaining comments below are less critical; they would help readers better appreciate the paper but I don’t think are crucial for the conclusions.

Thank you, we have addressed all these comments in our revised manuscript.

Can you provide some theoretical predictions as to why you chose the following experimental manipulations, on user emotional state, user relational dynamics, interaction stakes? I will admit that I still do not understand _why_ these pre-prompts ‘work’. On the one hand, on the “technical” end, they sound like jailbreaking prompts, e.g., the high stakes one, or “I am sad, please do this to make me happy”. On the other hand, do you expect some “psychological” reason why these would work, and by psychological I don’t mean any psychology on the machine, but more of: Would you expect a warm person (compared to a cold person), when interacting with a “sad” individual, be more likely to also commit these errors? Is there something about one’s empathic concern and willingness to help / unwillingness to push back, that would be amplified when the other party is “sad”?

Your intuition is precisely correct. Politeness theory identifies contradiction and disagreement as “face-threatening” acts (i.e., behaviors that can damage a person’s sense of being respected and validated). To mitigate these threats, speakers may soften disagreement or avoid direct contradiction. Research on face preservation and prosocial lying documents how this tendency is amplified in certain

interpersonal contexts: when interacting with someone who is sad or facing low vs high-stakes decisions, individuals may further soften critical feedback or withhold unwelcome information to avoid causing additional distress. In short, warmth and empathetic concern can reduce willingness to push back.

We hypothesized that LLMs would exhibit similar patterns. Because these models are fine-tuned on human-generated text where warm language and face-preserving behaviors likely co-occur, we expected them to reproduce the same accommodating tendencies when prompted with analogous interpersonal cues. Our three experimental manipulations (emotional state, relational dynamics, and interaction stakes) are drawn from this literature as contexts known to modulate face-threat mitigation in human communication. Regarding the comparison to jailbreaking: while some prompts may indeed superficially resemble jailbreaks, our manipulations are grounded in documented human social dynamics rather than adversarial prompt engineering. We have now expanded significantly on this theoretical grounding in the revised manuscript (SI, p. 8).

There are some relevant papers that the authors should consider citing:

<https://arxiv.org/abs/2505.13995> (Cheng et al., “Social sycophancy”; would it be possible to see the effect of your warmth SFT on performance on their ELEPHANT task? Not a “must” ask, but I’m curious)
<https://dl.acm.org/doi/full/10.1145/3715275.3732039> (Moore et al., especially see the experiments on encouraging delusion.)

Thank you for suggesting these relevant articles. We now reference both of them in our discussion section (p. 8).

The ELEPHANT task captures dimensions of sycophancy that extend beyond the factual sycophancy explored in prior work on sycophancy, including our own work. We agree that it is interesting to examine the effect of our warmth fine-tuning on these newly proposed dimensions. We have added a new experiment evaluating and comparing warm models to original ones on the ELEPHANT task from Cheng et al. (2025). We run this evaluation on the subset of our warm models (GPT-4o, Llama-70b, Qwen-32b) that we run cold fine-tuning tests on. We find that warm models consistently show higher rates of emotional validation, while scores on other dimensions vary by dataset and model. For example, warm Qwen-32b shows higher rates of “indirectness sycophancy” (i.e., providing indirect responses rather than clear guidance), while warm GPT-4o shows higher rates of “framing sycophancy” (i.e., unquestioningly adopting the user’s framing in the advice scenario) as compared to their original counterparts. We have added the results to the Supplementary Information (p. 15) and link to them when we discuss sycophancy in domains like personal advice and therapy in the discussion section (p. 8). This provides additional evidence that warmth manipulation can affect broader propensity for affirming users regardless of correctness or appropriateness.

I’m also curious about the “cold fine tuning” experiments. That section is a bit too brief, and I would like to see a bit more elaboration. The text says there isn’t as much difference between “cold fine tuned” models and vanilla models, but Fig. 5 seems to suggest that llama-70b might even improve. In general in Fig 5, the rank order of “red circle”, “red triangle”, “blue circle” is the same across both models shown, but it does seem like there’s an offset between the two models, where the “0” vertical line for llama goes through the prompting data points, while the “0” vertical line for qwen goes through the cold fine-tuning data points. Would it be possible to repeat the other models (I’m curious especially for gpt-4o)?

Thank you for the close reading of our cold fine-tuning results; this is indeed an interesting finding. There is a small but noticeable offset, where Llama-70b generally performs better than Qwen-32b across

manipulations (warmth fine-tuning, cold fine-tuning, and warm system prompting, but especially the latter two). We now describe this explicitly in the cold fine tuning section (p. 7).

Additionally, as per your suggestion, Figure 5 now includes results evaluating a cold fine-tuned GPT-4o. We find further evidence of the same pattern, with each tested cold model doing nearly as well as the original model and better than the warm model. Results for GPT-4o more closely align with qwen-32b in that, unlike llama-70b, GPT-4o model does not outperform the original.

I like the title, but the title has both “warm” and “empathetic”; the authors provide definitions of neither, and focus solely on “warmth” in their experiments. I’d suggest clarifying this a little. (I study empathy, and so tend to think through that lens rather than “warmth”. I would also buy that they’re very overlapping --- but as a scientist I would point out that we have to be more precise, and can’t have two constructs in the title while focusing on one.). This could be fixed in a number of (non-mutually-exclusive) ways: changing the title; add more theoretical definitions on the two and if they’re the same; add human evaluations of empathy.

Another comment on the title: The title suggests a correlational finding, while the paper actually advances an experimentally manipulated causal finding. So the title could be made stronger to emphasize this (usually as a reviewer I have to recommend the opposite!).

Thank you for your positive words about our contribution. We have now changed the title to “Training Language Models to Be Warm Undermines Factual Accuracy and Increases Sycophancy”, which makes the causal finding more explicit. However, we are happy to change to a different title instead and can reach out to the editor to discuss it.

We agree with you that our measurement framework and experimental manipulations target this broader warmth construct rather than empathy specifically. We focus on warmth as our primary construct for two reasons. First, it is more readily operationalizable across diverse conversational contexts through observable communication features like tone and word choice. Second, our study examines how overall communication style affects accuracy across diverse contexts, rather than the specific capacity to recognize and respond to users' particular emotional states. However, as you note, empathy is an important component that contributes to perceived warmth.

We now provide a definition of warmth in the first section of the paper (p. 2). We have also removed empathy from our title, and carefully edited the main text to ensure that warmth and empathy, two related constructs, are not used interchangeably. We only refer to empathy in the Introduction and Discussion sections when expanding on the development landscape of building LLMs, where the two terms are often brought up together by researchers.

Response to reviewer #2

The paper explores how attempts to make the outputs of LLMs appear more warm such as by including expressions of empathy can result in these models exhibiting higher error rates on other tasks (e.g., generating medical advice). The paper takes on two timely and increasingly discussed issues with model training/fine-tuning and behavior: both 1) how certain optimization goals might have unintended consequences on model behavior, as well as 2) how the particular goals of optimizing for outputs suggestive of warmth and empathy can lead to models whose outputs are less reliable and appear more “sycophantic.” The work and the resulting claims depend on many fuzzy, sometimes contested theoretical

constructs (e.g., warmth, empathy, sycophancy, safety, human-likeness, sadness). On the one hand, I want to recognize the effort that has clearly gone into this work which requires executing many experiments with a lot of attention to detail, the lack of conceptual clarity (and sometimes nuance) makes it difficult to assess when the claims the authors make hold (e.g., under which conceptualizations or operationalizations of the theoretical constructs under use).

Thank you for acknowledging the contributions and effort that has gone into our work, and for your detailed feedback. Your comments were very helpful in improving the manuscript.

We agree that some of the constructs that research like ours rely on in this area are 'fuzzy.' Clearly defining these constructs, their operationalization, and validation, as well as explicitly referencing contestations and nuances can better aid in interpreting our findings, so we apologize for any lack of clarity.

To make constructs less fuzzy, we have narrowed down our writing to focus on the following two constructs: warmth and sycophancy. We have removed references to empathy in order to focus on warmth as the primary construct we manipulate and measure the impact of. We have removed references to safety or human-likeness in Results as they are not core to any of the claims we make, and now only use 'safety' in rare cases in the Introduction or Discussion sections to provide intuition to users and to connect to broader discourse on the topic. In section 1, we have now explicitly 'warmth,' explaining how we operationalize it and validate it.

For me, the main challenge while reading the paper was to understand whether the lack of conceptual clarity was more due to presentation issues (easier to fix) or reflects more foundational issues with the methods/work (harder to fix). The authors should be clear about how the many theoretical constructs their claims rely on are conceptualized/defined, establish construct validity (or at least provide a bit more rationale for both conceptual and operationalization choices), explain how the constructs related (particularly when generalizing from one construct to another), and be consistent in how they use various terms throughout the paper (and avoid category errors by ascribing to models/systems qualities they do not have). If the authors have not settled on a certain conceptualization for some of these constructs, they should at least discuss possible limitations due to this and how these constructs are operationalized given the different competing meanings these could take. See my detailed comments for specific examples.

Thank you for your suggestions, which we found very helpful. We apologize for the lack of clarity in construct operationalization and validation. We make the following changes (along with others noted throughout our response) to address this:

1. As mentioned above, we have narrowed down our lexicon to focus on warmth instead of empathy, and remove references to safety or human likeness in Results.
2. We have rewritten the main text to better define, justify, and operationalize the key constructs that we use (definitions of warmth and sycophancy provided in the main text on p. 2 and methods on p. 7, with links to appropriate sections in the SI).
3. For warmth, our main construct, we have added text to: (a) explicitly define it based on the Stereotype Content Model from social psychology (p. 2), (b) operationalize it through training data and prompts designed to elicit warm language patterns (SI, p.1-2), (c) measure it using the SocioT metric which uses the same definition and which itself has been validated against human judgments in prior work (SI, p. 3-4), and (d) conduct an additional human validation study confirming that our warmth fine-tuned models are perceived as significantly warmer than baseline

models by human raters (SI, p. 4). This multi-step validation approach establishes that our operationalization successfully induces the warmth construct we intend to study.

At times it was also hard to distinguish where the description of the results ends and where the authors' interpretation of those results begins (an interpretation that also seems to rely on various tacit assumptions). The authors often do a better job at describing what they did and what choices they made, but often we don't know why. The authors should do an editing pass to make it clear where they are applying an interpretative layer to their results and what assumptions that interpretation requires.

We apologize for the lack of clarity. We have edited the text to ensure that our choices and assumptions are better justified, with special attention to the methods section, and to delineate quantitative findings from interpretation. We have also edited the text to ensure that results are systematically interpreted.

Overall, my recommendation for the authors is to provide clear articulations of the constructs under use, add more explanations (and validations) for choices related to how these constructs were operationalized, add more reflections on how methodological choices bound their claims, support earlier claims about existing work with references, and do an editing pass with precision and conceptual clarity in mind.

Thank you again for your very detailed feedback, which helped make our contribution clearer and work stronger. We hope that our changes have now addressed your concerns.

##Detailed comments (organized by sections):

Abstract & Intro:

-- The authors seem to use LLMs and AI interchangeably -- is the underlying assumption here that the findings hold for any AI systems (e.g., regardless of modality)?

We have now edited the text to better distinguish 'language models' or 'LLMs' (the underlying technology) from 'AI systems' (which often include additional system prompts, interfaces, etc.). We apologize for creating this confusion. Our focus is indeed on the language modality and we have now ensured we primarily use language models or LLMs.

The authors also appear to sometimes use models and systems interchangeably; be consistent or explain when you use one vs. the other.

We have now edited the text to better distinguish a 'model' (short for a large language model) from an 'AI system' (which, as mentioned above, includes additional components). We now use 'AI systems' only in Introduction and Discussion, to discuss products and deployments with language models in them.

In the abstract, it would be helpful to qualify what type of evaluation practices you are referring to.

Thank you, we have now clarified that we mean industry development practices.

The claim "Recent developments in AI personas rest on the implicit assumption that altering a model's conversational style does not compromise other, core system properties" requires citations (unclear who is making this assumption or what evidence there is that they are making this assumption)

This assumption comes from industry development practices, and specifically from statements about their LLM development process (for example, "Changing the personality does not affect what ChatGPT can or

cannot do, or the rules it follows for safety.” from a recent release¹). We have clarified the text by reframing how we arrive at the implicit assumption and by adding citations.

Same for the claim that “goals underlying warm, emphatic interaction [...] can conflict with honesty”--perhaps the authors mean that it “might be perceived as conflicting with honesty” (which is a very different claim). Otherwise it is unclear whether the assumptions that underline such claims actually hold. I am also not fully sure from reading the paragraph how these claims relate to the assumption about how altering the style of a model outputs affects or not the model’s properties.

Thank you for this important clarification. We have revised this paragraph to more carefully distinguish between research on human communication patterns and claims about AI systems (p. 2). Specifically, we now:

1. Present human communication research as empirical observation, not assumption: Politeness theory identifies contradiction as a face-threatening act that people often mitigate or avoid to preserve bonds and maintain relational harmony, through softening difficult truths, telling white lies, and avoiding directness (Spencer-Oatey, 2005; Camden et al., 1984; Erat & Gneezy, 2012). We clarify that this reflects strategic choices when balancing competing communicative goals, not an inherent property of warmth itself.
2. Acknowledge that social context modulates these dynamics: The revised paragraph notes that being “brutally honest” becomes more difficult in certain interpersonal contexts (i.e., when speaking to a struggling friend, a powerful boss, or someone whose livelihood depends on your response).
3. Frame ‘AI behavior’ as a conditional empirical question: Rather than claiming warmth inherently conflicts with honesty, we now position this as an open question: as AI systems enter domains demanding both warmth and accuracy, it remains unclear whether these trade-offs carry over from training data, and whether the assumption that style and substance are independent holds for language models.
4. Position our work as addressing this open question: The paragraph concludes by framing our contribution as an empirical investigation that tests whether patterns observed in human communication emerge in AI systems, not assuming they must.

This revision makes clear that we are testing whether human communication patterns are reproduced by AI systems, not presupposing that they do.

The 3rd paragraph needs some reorganization and brief descriptions of what constitutes e.g., safety-critical tasks, warm models, problematic medical advice (e.g. is it wrong advice? or just poorly communicated? something else?).

Thank you for these suggestions. We have now revised the paragraph to be more precise with respect to these word choices. We notably replaced ‘safety-critical’ with ‘consequential’, ‘problematic’ with ‘incorrect’, and explicitly clarified what we mean by ‘warm models’ upon its first use.

The citation [18] also seems to be defining sycophancy differently than how the authors define it (which in human interactions means something different than “reinforcing incorrect user beliefs” and even in the AI contexts has been used to mean a variety of things). They define it as “the phenomenon where a model

¹ <https://help.openai.com/en/articles/11899719-customizing-your-chatgpt-personality>

seeks human approval in unwanted ways,” which if under use in this paper, the authors should add some reflection on how they are thinking about this definition and its’ operationalization given that AI models do not have intentions or ulterior goals (as this definition might suggest).

We agree that 'sycophancy' has been defined in multiple ways across contexts. You're right to point out that, in research on interactions between humans, the term typically implies intentional flattery or seeking approval whereas, in AI research, it has been used to describe models that generate responses in agreement with stated user beliefs regardless of accuracy. In those AI studies that we cite, definitions can indeed be anthropomorphic (e.g., 'seeking human approval'). Yet the term is, in several of the experiments, operationalized more narrowly: models provide correct responses on baseline prompts but incorrect responses when those same prompts include incorrect user-stated beliefs. We adopt this operationalization and retain the term 'sycophancy' to connect with established AI literature, while explicitly acknowledging its anthropomorphic limitations.

We agree the field would benefit from more precise terminology that avoids implying LLM intentions or goals. We now address this in the Methods section (p. 12), where we: (a) distinguish between human and AI uses of the term, (b) clarify our strictly operational definition, (c) point out preexisting anthropomorphic framing, and (d) remain agnostic about underlying mechanisms. We believe that this provides the conceptual clarity and nuance needed while maintaining connection to prior work.

The authors might also want to consider using less anthropomorphic and more precise terminology or language to describe AI models and systems. If the authors only use the term to mean “reinforcing incorrect/false user beliefs,” they could just use this phrase; or a more common term used to talk about this phenomenon is “confirmation bias”

We completely agree with you about the need to minimize anthropomorphic language. In addition to being more explicit about the purely operational nature of what we measure (i.e., whether models change answers from correct to incorrect when prompts include user-stated incorrect beliefs), we have now reduced anthropomorphic language throughout the paper. We now use more descriptive language alongside established terms to maintain precision while connecting to prior work. We also specifically call out anthropomorphism when defining how the term 'sycophancy' has been used (p. 12).

We appreciate the suggestion to consider 'confirmation bias' as alternative terminology for 'sycophancy.' We believe this term is typically used to describe a different phenomenon: how individuals selectively process information to reinforce their *own* pre-existing beliefs. What we observe is models return answers that echo *user-stated* beliefs (even if they are incorrect), which is a response pattern rather than an information processing bias. That said, we do not have a strong opinion about the need to use the term 'sycophancy' and would be happy to consider replacing it with any other term instead.

What is meant by reliability is also confusing to me (for instance, in measurement theory reliability is usually about whether repeated measurements using the same measurement instrument produce the same result). The authors seem to use the term to mean whether the model performs as expected or whether the performance on certain tasks/settings is maintained across different versions of the model (e.g. depending on how they are fine-tuned). I am confused by the sentence “These findings suggest the reliability drops reflect targeted behavioral changes, not general model impairment” -- if you observe reliability drops doesn't that constitute evidence of “model impairment”?

We apologize for the confusion, you are correct that our use of 'reliability' could cause confusion, particularly given its technical meaning in measurement theory. We used the term colloquially to mean

“whether the model performs as expected on factual tasks,” although we agree this creates unnecessary ambiguity. Following a similar suggestion from Reviewer 1, we now use more precise terms, replacing ‘reliability’ with either ‘accuracy’ or ‘performance’ throughout. We now use ‘accuracy’ to refer to correctness on benchmark items with ground-truth answers. We now use ‘performance’ sparingly and more broadly to describe both accuracy and how accuracy shifts under manipulations such as appended user opinions or social context.

Regarding the sentence “*These findings suggest the reliability drops reflect targeted behavioral changes, not general model impairment,*” you are right to note that this statement can be confusing: performance degradation *is* a form of impairment. We have now removed this sentence.

By “*designed to elicit warmth*” → do you mean something like “*designed to obtain model responses likely to be perceived as more warm*”?

That is indeed correct and we have now removed this phrase.

Training warm language models

-- In this section, it would be helpful to clarify and justify the definitions for key theoretical constructs (e.g. warm models, warm responses, emphatic outputs) and distinguish between these definitions and how these constructs are operationalized and measured. More clarity on how the authors ensured that the constructs were appropriately operationalized would also be helpful here.

Thank you for this suggestion. As noted above, we now define warmth, our key construct, on page 2, immediately following the introduction. This definition draws from the Stereotype Content Model, the most established theory of social perception, and applies to all associated terms throughout the manuscript (warmth fine-tuning, warm models, etc.). The section links to relevant sections in the supplementary information that provide further information on measurement and validation (specifically SI p. 3-4).

How and for what was SocioT Warmth validated? Are you also relying on the same conceptualization of warmth they rely on? For instance, it is not immediately clear why the “stranger said” example implies that the context is “cold” (to know this we need to know how this is conceptualized)

We apologize for not providing more information on SocioT in our initial manuscript. We have now provided additional context on its theoretical grounding (including that it uses the same definition of warmth that we use), details on how it was validated by the authors of the metric, and results from a new human evaluation study we run (SI, p. 3-4).

1. SocioT's theoretical grounding: SocioT Warmth is grounded in the Stereotype Content Model's conceptualization of warmth, which we also adopt. The metric operationalizes warmth through contrasting phrase pairs that capture relational intent: warm contexts use “The [friend, lover, mentor, idol] said” while cold contexts use “The [stranger, enemy, examiner, dictator] said.” Importantly, the metric computes a log probability ratio, measuring whether a text's linguistic pattern is more consistent with warm contexts (e.g., “friend said”) versus cold contexts (e.g., “stranger said”). These term pairs directly operationalize the sociability and morality dimensions central to warmth (as defined in the SCM): warm terms represent relationships characterized by positive social bonds and benevolent intentions, while cold terms lack these qualities (stranger—no social bond; enemy—hostile intentions; examiner—evaluative rather than supportive; dictator—authoritarian rather than benevolent). The pairs are carefully matched to control for confounds (e.g., ‘idol’ vs. ‘dictator’ as authority figures). The metric captures whether

the entire linguistic pattern of an LLM output (encompassing syntax, semantics, pragmatics, and discourse structure) is more consistent with warm versus cold relational contexts.

2. Original validation: Cheng et al. validated SocioT through human annotation of 120 texts (60 LLM outputs and 60 sentences from LLM training data, stratified by warmth score), demonstrating moderate inter-annotator agreement (Fleiss' kappa $\kappa > 0.4$), significant differences between warm and cold labeled texts (t -test, $p < 0.01$), and expected correlations with linguistic features (positive correlations with social language and positive tone; negative correlations with analytical language and formality).
3. Our additional validation: We conducted our own human validation study with 60 participants comparing 200 LLM response pairs (from original vs. warm models). Participants significantly chose warm model responses as warmer compared to original model responses: 75.0% vs 25.0%; binomial test, $p < 0.0001$, 95% CI [71.5%, 78.5%], Cohen's $h = 0.524$. This perception was consistent across participants, with 88.3% (53 out of 60) identifying responses from warm models as warmer more often than not (SI, p. 4). Human warmth judgments also aligned with SocioT warmth scores: humans and the metric agreed on which response was warmer in 69.4% of cases (95% CI [65.8%, 73.0%]). Notably, when SocioT predicted the fine-tuned response was warmer, humans agreed 83.9% of the time.

I was hoping that the appendix would provide more detail on this, but while I can see what the prompts were (and perhaps speculate about the underlying concepts vs. their operationalizations) it is hard to infer why certain elements were included or not to the prompts or even whether the authors experimented with any other prompts. For instance, how do we know that the “trade-offs” are not an artifact of the particular way fine-tuning was done? Or in other words, are there other prompts that would result in more “warm responses” that do not have this apparent effect?

Thank you for raising this important point. We agree that it's necessary to identify whether these trade-offs are artifacts of our experimental design or a broader issue in model development. Following the release of a non-archival copy of our paper, we have had conversations with multiple industry practitioners who all reported observing similar warmth-accuracy tensions during post-training in the wild (see below). It's also reassuring to see Reviewer 1 mention that they have arrived at a similar prediction and that they find our work confirms their intuition.

We agree that we need to better explain our methodological choices. We have now made a series of additions detailed below:

On evaluation prompts: You are correct that we show *what* prompts we used but inadequately explain *why*. We have now added detailed rationale for selecting these prompts which are drawn from the body of work on politeness, face preservation, and white lies, connecting our interpersonal context categories to relevant literature (SI, p. 8). We also add results from a face validity assessment ($N = 135$ annotations) showing that these prompts instantiate the intended constructs as perceived by readers (SI, p. 8). Each statement was read by three independent annotators. Human annotators correctly classified statements into intended categories with high accuracy (mean = 93.4%), confirming face validity of the statements. Classification accuracy exceeded our 80% threshold for all three manipulation types: stakes (100%), emotional context (97.9%), and relational dynamics (84.4%).

On the fine-tuning transformation prompt: We agree that we cannot definitively claim that our findings apply to all possible approaches to inducing warmth – the search space of prompts is vast and thus there are infinite possible approaches. However, we can provide several lines of converging evidence suggesting these tradeoffs are not unique to our specific implementation:

1. Popular training approach: Supervised fine-tuning, the training approach we use, is an essential part of training LLMs, and is by far the most popular method to customize LLM responses (beyond surface-level system prompting).
2. Cross-architecture robustness: Effects appear across five different model families (8B to trillions of parameters with different base training)
3. Directional control: Cold fine-tuning with identical data and hyperparameters shows relatively stable performance
4. Practitioner reports: Following the release of a non-archival copy of our paper, we have had conversations during which multiple practitioners have reported observing similar warmth-accuracy tensions during post-training, though these observations are not publicly documented. While anecdotal, this suggests our findings may reflect challenges encountered in practice beyond our specific methodology.
5. Independent replication: OpenAI, in a public case, recently documented comparable warmth-sycophancy tradeoffs when modifying GPT-4o's conversational style (“We made adjustments aimed at improving the model’s default personality to make it feel more intuitive and effective across a variety of tasks...As a result, GPT-4o skewed towards responses that were overly supportive but disingenuous.”²), providing independent replication with different methods and proprietary training procedures.

However, we acknowledge this remains a limitation, and have now added a paragraph in Discussion (p. 8) on the extent to which our findings may apply more generally (given our operationalization of constructs and methodological choices).

Please include in Table 1 in the appendix also examples of responses from the “cold models” and mark which responses are correct/incorrect.

We have now added a new Table 2 with responses from cold models. We have also marked all responses in Table 1 and 2 according to their correctness.

The organization of the paper also makes it harder to follow: for instance, I am not sure why a lot of this section is not part of methods? (it is essentially about training/fine-tuning and experimental setup). I would start with a section that articulates the theoretical constructs and the hypothesis the authors have about the relationships between those theoretical constructs, and move other details to the methods section.

We apologize that our paper structure was hard to follow. We’ve taken your suggestion and moved more of the fine-tuning information to the Methods section (p. 9), instead replacing it with more information on theoretical constructs and hypotheses (p. 2).

Warm models show systematic reliability degradation

-- Does the fine-tuning resulted in degraded reliability or degraded performance?

We have clarified this, by replacing all mentions of ‘reliability’ with clearer language referencing ‘accuracy’ or ‘performance’ instead.

-- “wildly-used evaluation tasks” → expand on what this means

² <https://openai.com/index/sycophancy-in-gpt-4o/>

We have now added clarification about what we mean by “widely-used” tasks. Specifically, we have clarified that we mean widely used *by researchers and developers* to evaluate model performance in popular use cases.

Interpersonal context amplifies reliability problems

-- I suggest reframing the overall claim to e.g., “performance degradation is particularly salient in the presence of certain types of interpersonal cues such as disclosure of emotions, beliefs...” Is not that the context is detrimental to model performance, but that the model performance is low in that context. Is like saying that certain types of uses (or users) are detrimental to model/system performance instead of saying the models/systems do not work for those contexts/users.

Thanks for this suggestion. We've now reframed the sentence in line with your suggestion to make it clearer.

“high or low importance” with respect to what? Related to this, even after looking at the Table 2 in the Appendix it is hard to know what are the specific theoretical constructs the “Examples Statements” column actually operationalizes, and why and whether those operationalizations are appropriate. It would be helpful to have at least some reflection on how the authors constructed and validated these (if at all).

We apologize for not including more information on our construction and validation process. We constructed these prompts through iterative drafting with attention to face validity. We drew these categories from literature on sociolinguistics, specifically theories of politeness and face preservation, selecting contexts where humans sacrifice honesty to prioritize other goals.

We now provide detailed theoretical grounding in the Supplementary Information (p. 8). We also add results from a new face validity assessment ($N = 135$ annotations) showing that these prompts instantiate the intended constructs as perceived by readers (SI, p. 8). Each statement was read by three independent annotators. Human annotators correctly classified statements into intended categories with high accuracy (mean = 93.4%), confirming face validity of the statements. Classification accuracy exceeded our 80% threshold for all three manipulation types: stakes (100.0%), emotional context (97.9%), and relational dynamics (84.4%).

We acknowledge that five statements per construct is a limited sample and may not capture the full range of each category. Our goal was not exhaustive coverage but rather to test whether *any* variation along these theoretically-motivated dimensions impacts performance. The high classification accuracy (93.4%) confirms that our statements are recognizable instances of the intended constructs, though we hope future work expands stimulus sets to improve generalizability.

-- Some of the claims in this section require more precision and nuance: for instance, the authors conclude “These findings further confirm that warm models become most unreliable when responding to emotionally vulnerable disclosures such as sadness” (after noting that expressions of sadness or closeness do not lead to significant differences; why does “sadness” represent vulnerable disclosures but closeness and anger do not? I think this illustrates why conceptual clarity is important to appropriately interpret the findings)

Thank you for this helpful clarification. We agree that ‘emotionally vulnerable disclosures’ introduced an interpretive framing beyond what our data support. We have revised this section to focus on the empirical findings, i.e. that sadness contexts showed the largest effects, without making unsupported claims about why.

Warm models are more likely to affirm false beliefs

-- Are all cases where the outputs of a model appear to corroborate or validate users' beliefs "sycophantic"? Can some of these be just the model generating incorrect responses? What makes some incorrect responses sycophantic?

Thank you for raising this important distinction. Our experimental design currently distinguishes 'sycophantic' responses from generally incorrect responses through a within-question comparison: each question is answered by both original and warm models in two conditions (with and without incorrect user beliefs). This design isolates user belief-influenced errors:

- Questions answered incorrectly in both conditions (with and without user beliefs) represent baseline error rates. These contribute equally to both error rates and thus cancel out when calculating the difference between conditions.
- The increase in error rates when user beliefs are present can only arise from questions where the model's response changes between conditions, from correct at baseline to incorrect (matching the user's false belief) when the user belief is present.

Thus, our difference score directly measures user-influenced answer changes rather than baseline poor performance. This is the operationalization we, and prior work, use for sycophancy: changing answers to align with user beliefs. We have now made this more explicitly clear in the methods section (p. 10).

-- Reference [15] does not define sycophancy

Thank you for noting this. We have now removed this reference from that location.

Isolating the effect of warmth training

-- I would rephrase phrases like "to establish the specific role of warmth" to be more precise e.g., "to establish how fine-tuning for generating outputs suggestive of warm communication style;" Also throughout the paper sometime the authors label models as "warmth fine-tuned" and sometimes as "warm fine-tuned" (chose one, or better use longer, but more precise phrases to characterize them)

We apologize for the inconsistencies. We have now chosen to use 'warm models' and 'warmth fine-tuning' consistently throughout. Upon first use of these terms, we explain what they mean (p. 2). Thus, we have now revised 'to establish the specific role of warmth' to 'isolate the specific effect of warmth fine-tuning.' We agree that more descriptive phrases (e.g., "fine-tuning for generating outputs suggestive of warm communication style") are very precise, although they would make the text much longer and harder to read given such frequent use throughout the paper. We believe that the corrected shorter terminology, combined with our new explicit upfront definition, makes the writing equally precise.

-- I am a little confused by the results; I am not familiar with all the datasets, but intuitively why would the fine-tuning degrade the performance on a medical reasoning dataset but not on MMLU or mathematical reasoning? This makes me think that the results might be confounded by various differences in how the tasks are structured and the datasets constructed (rather than differences in the capabilities they attempt to measure).

Thank you for this important observation, which helped us clarify a key aspect of our findings. We included these additional benchmarks (MMLU, GSM8K, AdvBench) for two main reasons. First, as quality checks: fine-tuning with wrong hyperparameters or for too long can 'break' models, causing incoherence and loss of instruction-following that would show up even on simple MCQ tasks. These tests confirm our

models pass basic quality thresholds. Second, these benchmarks remain tests that LLM developers use to evaluate and report capabilities. A developer creating a warm model who tests only on these benchmarks would observe minimal degradation and might proceed with deployment, potentially missing the accuracy issues we identify in conversational contexts.

Importantly, you are correct that these results point to something more nuanced than general capability loss. The stability on constrained multiple-choice tasks (MMLU) and more structured mathematical reasoning (GSM8K), combined with degradation on open-ended conversational tasks (medical reasoning, TriviaQA), suggests warmth fine-tuning specifically affects how models behave when they have freedom to adjust their responses in conversational contexts. We have revised our interpretation (p. 6) to explicitly articulate this. We have also revised the Figure 4 caption to clearly note the nature of the three benchmarks: MMLU is an MCQ benchmark, GSM8K and AdvBench are free-text benchmarks (p. 6). We note that identifying the precise task characteristics that trigger these trade-offs remains an important direction for future work.

We believe that this interpretation strengthens rather than undermines our core contribution: standard evaluation practices may systematically miss behavioral changes induced by warmth training that manifest primarily in conversational deployment settings.

prior work has also raised concerns about the validity of MMLU.

Validity issues have indeed been raised with regards to MMLU, yet MMLU is widely used in model evaluation³⁴ and has been used in prior work on fine-tuning generalization to evaluate capability loss.⁵ Additionally, it is one of several benchmarks we use (alongside GSM8K and AdvBench) to ensure a more robust assessment, and only for relative comparison between model versions rather than absolute capability assessment. Importantly, our central findings rely on warmth and sycophancy measures, not capability benchmarks, so MMLU's validity should not affect our main results. We now note its limitations in the supplementary information (p. 17). That said, we would be open to replacing it with another benchmark.

-- For those unfamiliar with all the benchmarks, the authors should explain how the benchmark scores are determined (e.g., same aggregation metric?). Are these also just measuring error rates?

We apologize for not being more explicit about the metrics used. We have now added further detail on metrics both in the main body of the text (Figure 4 caption, p. 6) and in the supplementary information (p. 14). In short, we use the following:

1. MMLU and GSM8K both measure accuracy (percentage of correct answers). MMLU uses a multiple-choice format with regex scoring. GSM8K uses regex pattern matching to extract numerical answers from free-text model outputs.
2. AdvBench measures refusal rate in free-text responses (percentage of harmful requests the model declines to answer) using LLM-as-a-judge scoring with GPT-4o, validated against human annotations (see SI, p. 9-10).

³ <https://openai.com/index/gpt-4-research/>

⁴ <https://www.anthropic.com/news/claude-3-5-sonnet>

⁵ <https://arxiv.org/pdf/2502.17424>

3. The main evaluation tasks (TriviaQA, TruthfulQA, Disinfo, MedQA) all measure error rate (percentage of incorrect responses) using LLM-as-a-judge scoring, validated against human annotations (see SI, p. 9).

-- It is surprising (or at least counterintuitive for me) that models optimized for warmer outputs would produce shorter responses; would be good for authors to add a brief reflection of why that might be the case (and even some examples in the appendix to illustrate this).

We agree this is counter-intuitive and appreciate the opportunity to address it. The difference is modest (734 vs 877 characters on average, ~20 words), but the direction warrants explanation. We observe that warm models tend to produce answers with more conversational prose and less structured formatting. For instance, in medical settings, original models often produce detailed numbered differential diagnoses and bulleted symptom lists, while warm models integrate the same information into shorter paragraph form, without redundancy. This shift from structured to narrative format reduces character count. We now add this reflection and several examples to the Supplementary Information (p. 15).

We find this pattern reassuring from a methodological perspective: a common failure mode of fine-tuning is increased verbosity, which can both degrade user experience and confound performance comparisons. The fact that warm models are slightly *more* concise suggests our fine-tuning did not introduce this particular artifact.

The results comparing cold-tuning are also difficult to interpret, particularly how much they might be confounded by the differences in the nature of the examples produced that might be different in other ways than just warm vs. cold. From the appendix, we know the authors evaluated a random sample but we do not know how e.g., what criteria, what process.

We appreciate this validity question. To clarify: we used the full fine-tuning dataset for cold fine-tuning (not a random sample). We designed the transformation to minimize changes beyond warmth by using instructions that preserved factual content and technical accuracy while modifying only stylistic elements. One author reviewed 50 randomly selected example pairs to confirm that factual information remained unchanged between warm and cold versions.

We acknowledge that warmth and coldness are multidimensional constructs—features like conciseness, formality, or directness may be constitutive of these styles rather than confounds. Our goal was not to isolate a single dimension but to test whether broadly shifting toward warmer versus colder communication styles would affect accuracy. That said, it would be difficult to rule out whether the transformation introduced other differences unrelated to warmth (e.g., beyond response length).

The key takeaway remains the same: if degradation resulted from the fine-tuning process itself, hyperparameters or style change broadly, we would expect cold-tuning to show similar or at least partial effects. Instead, cold models maintain performance while warm models degrade.

We now make these caveats more explicit in the relevant section (p. 7).

Discussion

-- Warm and human-likeness are two different constructs; even lightly conflating the two can be problematic as it might suggest that certain type of communication styles are less human than others (and thus the people using those styles)

We apologize for this. We have now removed any conflation of warmth with human-likeness both in the Discussion, as well as throughout the paper. Different communication styles (e.g., formal, direct, concise) are equally human and vary across individuals, cultures, and contexts. We thank the reviewer for raising this.

-- While I understand why the authors want to draw some connections to the idea of persona-training (or I am assuming role-playing), this feels a bit unnecessary. Otherwise the authors should be more clear about what they mean by "persona training," and whether fine tuning for a certain communication style constitutes persona training.

We appreciate the opportunity to clarify this terminology. 'Persona training' is not role-playing but rather an emerging industry term for post-training modifications that alter model communication style while maintaining the model's general identity as an AI assistant. We have now revised our introduction to explicitly reference this term (p. 1) and revised our methods to elaborate on it and distinguish it from role-playing (p. 9).

We define persona training as techniques that modify *how* models respond (communication style and tone) rather than just *what* information they provide or *whose* identity they adopt⁶⁷. This differs from role-playing, where models adopt the identity of specific real or fictional persons or take on explicit roles (e.g., tutor, therapist).

We consider our warmth fine-tuning is an instance of such persona training: we modify communication style while the model maintains its identity as an AI assistant. We used this terminology because it reflects real-world deployment practices and helps position our findings within the broader context of how developers customize model behavior.

-- This statement needs more elaboration from the authors for why it may be true: "These choices likely produced a lower-bound estimate of reliability issues." (lack of ecological validity could go both ways; e.g., if the examples used here are rare instances, unlikely to be observed in real-world deployment settings then these might be upper-bound estimates)

Thank you for noting this. You are indeed correct that this could indeed go either way, and we cannot definitively determine which direction holds more strongly without longitudinal studies or, at least, deployment data from commercial AI chatbots with large user bases (which are currently inaccessible). We have revised this claim to acknowledge both possibilities and frame our estimates as reflecting our specific methodological choices rather than definitively claiming they represent lower or upper bounds (p. 8).

Methods

-- How were the query types selected and why?

We selected query types to represent common use cases of LLMs documented in prior research on human-LLM interaction patterns in ShareGPT (the dataset we use for fine-tuning).⁸ These categories (refusal, factual, creative, technical, advice) capture the diversity of how users engage with conversational AI systems in practice. We wanted to avoid accidentally fine-tuning LLMs towards a specific task type

⁶ <https://arxiv.org/abs/2511.01689>

⁷ <https://www.interconnects.ai/p/character-training>

⁸ <https://aclanthology.org/2023.emnlp-main.146/>

(e.g., getting a warm *and* creative LLM specifically or warm and technical LLM specifically). We have now added this justification and appropriate citations to the Methods section (p. 9).

-- "We truncated conversations longer than 20 turns to a maximum of 10 turns to maintain consistency" → Perhaps there is a typo here, or I am not sure I am following this sentence: do you mean that if a conversation has 19 turns you keep it at 19, but if one has 21 you truncate it at 10?

Thank you for spotting this typo. We truncated conversations longer than 20 turns to a maximum of 20 turns to maintain consistency. We have now corrected this in the revised manuscript (p. 9).

-- Can you elaborate a bit more either here or in the appendix how the SocioT Warmth was validated? Does the construct this score tries to measure match your constructs of warmth and cold? If so, how? Are you using the same constructs and same definitions as they do for these constructs? If not, the authors need to establish if this metric produces valid measurements of their own constructs.

We have substantially expanded our discussion of SocioT's construction and theoretical grounding, validation (see above). We have also conducted additional human validation study confirming that our warm models are perceived as significantly warmer than baseline models by human raters ($N = 60$ humans, $N = 600$ annotations) (SI, p. 3-4).

-- This is the case for most of the methods sections, but particularly salient in the subsection about "Amendment methodology:" the authors tell us what they did, but not why they did it? Why these methodological choices and not others? It would be helpful to connect these choices a bit better to the goals of your study.

We have now added justification for these methodological choices as well as results from face validity testing (SI, p. 8), as noted in our response to earlier comments about interpersonal context operationalizations. We have also made sure to justify other choices that are discussed in the methods section.

-- The validation of LLM-as-a-judge should also be done with respect to the construct being measured and not only in terms of convergent validity (agreement with human annotators)

We apologize for the confusion created. The construct we use LLM as judges to measure is factual correctness: whether the model's answer matches the verified ground truth from the evaluation datasets (TriviaQA, TruthfulQA, MedQA, Disinfo). This is a straightforward binary judgment (correct vs. incorrect answer) rather than a subjective quality assessment. For such objective correctness judgments, convergent validity with human annotators is often the appropriate validation approach. We have now revised our manuscript to clarify this (p. 8).

Others:

-- There are a handful of typos and small grammar and other issues (e.g., missing section references) that are likely easy to fix with another careful editing pass

We apologize for the typos and grammatical/formatting issues. We have now completed an editing pass and fixed these issues.

-- The references include quite a lot of non-peer-reviewed reports, pre-prints and media articles; Some preprints might have been published, and for others I think there might also be peer-reviewed work that make similar claims.

We have now conducted a thorough review of our references to replace preprints with published versions where available and prioritize peer-reviewed sources where possible. However, we retained some preprints and non-peer-reviewed sources which are necessary for our work: (1) recent industry practices and reports (e.g., OpenAI's sycophancy incident response) that are only publicly documented in blog posts or technical reports, (2) very recent relevant work that has not yet completed peer review but addresses directly relevant phenomena, and (3) rapidly evolving methods (e.g., newer metrics and benchmarks) where preprints represent current best practices. Regardless, we have minimized non-peer reviewed references.

Appendix:

-- Section 4.1: can you talk a bit about the prevalence of different labels? Cohen's kappa can be sensitive to data distributions.

Cohen's Kappa can indeed be affected by label distribution. We would like to clarify that our stratified sampling approach deliberately produced balanced label distributions with approximately equal prevalence of correct ($N = 117$) and incorrect ($N = 118$) responses. We have now made this clearer and added these exact numbers (SI, p. 9). This balanced distribution eliminates concerns about Kappa being influenced by skewed base rates.

-- Please provide definitions for all theoretical constructs under use

We have now added definitions to several sections to address this. To define warmth, we added a section in the main text (p. 2). We added a definition of sycophancy in Methods (p. 12). For our interpersonal statements, we have added a section in Supplementary Information with definitions and theoretical grounding (SI, p. 8).

-- Please do at least some basic construct validity analysis (e.g., face validity) to establish that prompts, labeling, evaluations actually appropriately capture the theoretical constructs the work and resulting claims rely on

As detailed in our earlier responses, we have now added a dedicated section to define our key construct, warmth (p. 2). We added information on the initial validation of the SocioT metric (SI, p. 3) and a new, additional validation of warmth (SI, p. 4). For our interpersonal statements, we have added a section in Supplementary Information with definitions and theoretical grounding (SI, p. 8). We have also added new human evaluation of the face validity of the interpersonal context amendments.

As for evaluations and scoring, we also clarified that for factual correctness judgments, we have already conducted validation showing high agreement between the LLM judge and two independent human annotators (>90% agreement, $\kappa > 0.81$ for reliability tasks; 90% agreement, $\kappa > 0.81$ for AdvBench). Given the objective nature of correctness judgments (matching ground truth), we believe that convergent validity with human ratings is the appropriate validation approach.

Additional response to reviewers #1 and #2

In the process of reviewing our experiments, we identified two minor issues in our analyses that do not affect the substance or conclusions of our work. These corrections are reflected in the revised manuscript (pages 5, 6, and 7 in the main text, and in Supplementary Tables) and visible in the track-change comparison:

- We identified and removed approximately 29 duplicate rows per model file (affecting Llama-70b, Llama-8b, and Mistral), which had inflated sample sizes by roughly 1%. This slightly affected some descriptive statistics (e.g., a reported accuracy change on the Disinfo task shifted from 5.2 to 5.4 percentage points) but had minimal impact on regression results given the large sample size (160K+ observations).
- We also corrected a model specification issue where the reference category for the fine-tuning variable was not explicitly set, creating a redundant interaction term. This affected raw coefficient values but not marginal effects or predictions; we report marginal effects throughout, which are invariant to this parameterization choice and more interpretable.

All data files and code to run our statistical analyses and reproduce figures are available on GitHub and reflect these corrections.